# Unlocking Post-hoc Dataset Inference with Synthetic Data

**Bihe Zhao** [1]  **Pratyush Maini** [2][3]  **Franziska Boenisch** [1]  **Adam Dziedzic** [1]

## Abstract

The remarkable capabilities of Large Language Models (LLMs) can be mainly attributed to their massive training datasets, which are often scraped from the internet without respecting data owners' intellectual property rights. Dataset Inference (DI) offers a potential remedy by identifying whether a suspect dataset was used in training, thereby enabling data owners to verify unauthorized use. However, existing DI methods require a private set—known to be absent from training—that closely matches the compromised dataset's distribution. Such in-distribution, held-out data is rarely available in practice, severely limiting the applicability of DI. In this work, we address this challenge by synthetically generating the required held-out set. Our approach tackles two key obstacles: (1) creating high-quality, diverse synthetic data that accurately reflects the original distribution, which we achieve via a data generator trained on a carefully designed suffix-based completion task, and (2) bridging likelihood gaps between real and synthetic data, which is realized through post-hoc calibration. Extensive experiments on diverse text datasets show that using our generated data as a held-out set enables DI to detect the original training sets with high confidence, while maintaining a low false positive rate. This result empowers copyright owners to make legitimate claims on data usage and demonstrates our method's reliability for real-world litigations. Our code is available at https://github.com/sprintml/PostHocDatasetInference.

## 1. Introduction

Large language models (LLMs) have recently achieved remarkable success in a broad range of tasks, fueled by the availability of massive high-quality text corpora often scraped from the internet (Weber et al., 2024; Penedo et al., 2024). While this practice enables LLMs to generate high-quality text and to excel on benchmarks, it also raises serious concerns related to intellectual property rights (Reuters, 2023; Gry, 2023; Sil, 2023), data privacy (Duan et al., 2023a;b; Hanke et al., 2024; Hayes et al., 2025), and transparency (Rahman & Santacana, 2023; Wu et al., 2023). The reliance on potentially unauthorized data creates an urgent need for methods that allow independent authors to verify whether a given dataset has been used to train an LLM without the explicit consent of the model provider.

A promising approach to addressing these concerns is *dataset inference* (DI) (Maini et al., 2021; Dziedzic et al., 2022a;b; Maini et al., 2024; Dubiński et al., 2025; Kowalczuk et al., 2025), which aims to determine whether a suspect dataset has contributed to a model's training. This puts power in the hands of data owners to monitor and exercise their intellectual property rights. Despite its potential, DI currently faces a critical bottleneck: it requires a held-out set—a dataset known to be absent from training—that shares the same distribution as the suspect dataset (Zhang et al., 2024a). In practice, however, such an in-distribution held-out set is rarely available. Data creators do not typically reserve a dedicated held-out set for legal or auditing purposes, and any disclosed held-out data could itself be repurposed for future training, further complicating the verification process. Moreover, even when a dataset owner can provide some held-out samples, any slight distributional discrepancy from the original suspect data can undermine DI by inflating false positives (Das et al., 2024; Duan et al., 2024; Meeus et al., 2024; Maini & Suri, 2024).

To illustrate the brittleness of using seemingly IID (Independent and Identically Distributed) held-out data, we demonstrate in Section 3 that even in a simple scenario—where an LLM is fine-tuned on blog posts from a *single* author—there exists a distributional shift between training data (members) and randomly held-out blog posts from the same author. This highlights how even subtle variations in held-out data can undermine DI. Malicious actors may exploit this vul-

---
[1]CISPA Helmholtz Center for Information Security [2]Carnegie Mellon University [3]DatologyAI. Correspondence to: Bihe Zhao <bihe.zhao@cispa.de>, Pratyush Maini <pratyushmaini@cmu.edu>, Franziska Boenisch <boenisch@cispa.de>, Adam Dziedzic <adam.dziedzic@cispa.de>.

*Proceedings of the 42nd International Conference on Machine Learning*, Vancouver, Canada. PMLR 267, 2025. Copyright 2025 by the author(s).

*Our Work*

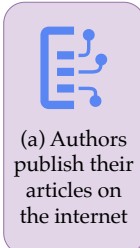 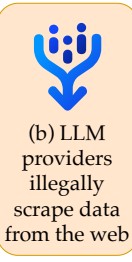 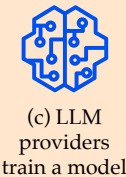 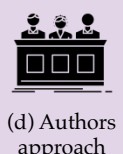 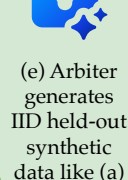 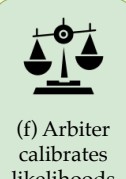 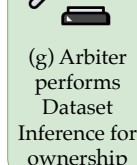

| (a) Authors publish their articles on the internet | (b) LLM providers illegally scrape data from the web | (c) LLM providers train a model on this data & deploy it | (d) Authors approach arbiter with copyright claim | (e) Arbiter generates IID held-out synthetic data like (a) | (f) Arbiter calibrates likelihoods for synthetic & real data | (g) Arbiter performs Dataset Inference for ownership |

*Figure 1.* **Dataset Inference Procedure with Synthetic Held-Out Data.** This figure presents a high-level overview of how the proposed dataset inference (DI) process will take place in real-world use cases. **(a-d)** LLM providers scrape proprietary author data from the internet, and train an LLM on it. Authors who suspect unauthorized use may approach an arbiter with a copyright claim. To resolve such a dispute, the arbiter must perform dataset inference **(g)**. However, this requires the presence of a held-out dataset that is IID to the suspect set. **Our Work: (e)** The arbiter generates IID synthetic held-out data that mimics the author's original data. **(f)** The arbiter calibrates likelihoods between real and synthetic data to ensure fair comparison, enabling them to reliably perform dataset inference.

nerability by strategically introducing *shifted* held-out data, falsely accusing model owners of copyright infringement and further reducing the reliability of DI methods.

In this work, we address these challenges by proposing to *synthetically generate* held-out data for DI, bypassing the need for in-distribution held-out data. This vision, however, is non-trivial to achieve. First, the generated texts must be realistic, high-quality, and sufficiently diverse to approximate the distribution of the original data. Second, the generation process itself may introduce a distribution shift between natural and synthetic held-out data. Such a shift complicates DI: if a difference is observed between the suspect and held-out sets, it becomes unclear whether this difference arises from a genuine membership signal (*i.e.,* the target model behaves differently on the suspect data because it has seen it during training) or merely from the distribution shift (*i.e.,* the model behaves differently on suspect data because it is natural data). Recent studies have extensively highlighted this issue in the context of Membership Inference Attacks (MIAs) (Shokri et al., 2017), where distribution shifts lead to misleading evaluation results (Das et al., 2024; Zhang et al., 2024a; Maini et al., 2024; Dubiński et al., 2025).

To this end, we first train a carefully designed text generator on the suspect dataset itself, on a suffix completion task (Section 4.1). This approach produces high-quality datasets with only a small distributional shift from the suspect texts. However, even small shifts in distribution can undermine DI's reliability. To address this, we introduce a *post-hoc* calibration step (Section 4.2) to ensure that the generated held-out set can serve as a reliable reference for DI. Specifically, we disentangle the effects of distributional shifts from the actual membership signal—a critical factor in DI. To achieve this, we propose a dual-classifier approach: (1) A *text-only classifier*, trained to distinguish natural (original) from generated data. (2) A *membership-aware classifier*,

which incorporates both the textual features and DI's standard membership indicators (e.g., perplexity, min-k probabilities). The key insight is that any performance advantage of the membership-aware classifier over the text-only classifier must arise from the presence of membership signals rather than distributional artifacts. This difference serves as our DI signal, allowing us to infer whether the suspect dataset was used in the target model's training. This calibration strategy enhances DI's robustness, reducing false positives while maintaining high detection accuracy.

We demonstrate the effectiveness of our approach on diverse textual datasets, ranging from single-author datasets (e.g., personal blog posts) to large-scale, multi-author collections such as Wikipedia. Our results show that using *synthetic* held-out data, combined with calibration, enables DI to detect unauthorized training data use with high confidence while keeping false positives low. This expands the practical applicability of DI and provides a pathway for data owners to safeguard their intellectual property in an era of LLMs.

## 2. Background and Related Work

### 2.1. Membership Inference

MIAs focus on deciding if a single data point was included in a given model's training dataset and often serve as features extractors for DI. In the LLM domain, MIAs exploit different signals to distinguish between members (training data points) and non-members (data points not used during training). For instance, LOSS exploits the perplexity or loss function of the target model (Yeom et al., 2018). Shi et al. (2024) find that the rare words in a sequence can leak more privacy information, and select K% tokens with the smallest probabilities for evaluation. Min-K%++ further improves upon the Min-K% approach by introducing two calibration factors (Zhang et al., 2024b). Zlib ratio (Carlini

et al., 2021) uses the compression rate of z-library to normalize the perplexity of the target model. Neighborhood-based methods compare a suspect sequence with its neighboring texts, which can be produced by synonym substitution (Mattern et al., 2023) or paraphrasing (Duarte et al., 2024). Moreover, reference-based methods compare the output signals on a suspect sample between the target model and a reference model (Fu et al., 2024). Yet, many recent works have shown that the evaluation of MIAs suffers from a falsified experimental setup, where a distributional shift exists between the member and non-member sets (Zhang et al., 2024a; Maini et al., 2024; Das et al., 2024). Duan et al. (2024) show that most MIAs only perform slightly better than random guessing if evaluated correctly on non-biased benchmarks. Recently, Kazmi et al. (2024) proposed how to de-bias MIAs from this distribution shift—which we use as a foundation for our DI calibration.

## 2.2. Dataset Inference

To strengthen the signal from training data further beyond MIAs, Maini et al. (2021) introduced DI. DI aggregates the membership signal over multiple data points, often referred to as *suspect set*, to decide whether a given model was trained on this data. More formally, given a target model $f$, DI aims to detect whether $f$ was trained on the suspect dataset $\mathcal{D}_{\text{sus}}$. Therefore, it needs an additional held-out dataset $\mathcal{D}_{\text{val}}$ from the same distribution as $\mathcal{D}_{\text{sus}}$. Given both sets, DI extracts membership features from the data points in $\mathcal{D}_{\text{sus}}$ and $\mathcal{D}_{\text{val}}$, aggregates all features per given sample, and then scores these aggregate features through a scoring model. The scores should be lower for members than for non-members. Then, DI performs statistical hypothesis testing on the scores of $\mathcal{D}_{\text{sus}}$ and $\mathcal{D}_{\text{val}}$. The null hypothesis is that the average scores for $\mathcal{D}_{\text{sus}}$ are higher than or equal to the scores for $\mathcal{D}_{\text{val}}$. If the statistical test manages to reject this null hypothesis, this is a confident indicator that the data points from $\mathcal{D}_{\text{sus}}$ are indeed members of model $f$'s training data. Otherwise, the test is considered inconclusive.

How to extract the best membership features from the data points varies based on the learning paradigm. For example, the original DI for supervised classification models (Maini et al., 2021) designs a random walk strategy to estimate the distance between data points and the decision boundary of a supervised model. For self-supervised models, Dziedzic et al. (2022b) use Gaussian Mixture Model to estimate the representational differences between the training dataset (members) and the test data. Recent dataset inference methods for LLMs (Maini et al., 2024), Diffusion Models (Dubiński et al., 2025), and Image Autoregressive Models (Kowalczuk et al., 2025) build on existing membership inference attacks (MIAs) tailored to each type of generative model. These methods extract membership-related features using the appropriate MIA and then apply a linear model to

*Table 1.* **The distributional shift (GPT2 AUC) and DI p-value between a suspect set that consists of *non-members* and held-out blog posts.** Here, p-value $< 0.05$ indicates DI incorrectly suggests that the suspect set is a member set.

| Sequences per Blog | 5 | 10 | 15 | 20 | 25 |
|---|---|---|---|---|---|
| GPT2 AUC (%) | 52.0 | 55.2 | 53.2 | 58.2 | 58.6 |
| DI p-value | 0.002 | <0.001 | <0.001 | <0.001 | <0.001 |
| True Membership | ✗ | ✗ | ✗ | ✗ | ✗ |
| Inferred Membership | ✓ | ✓ | ✓ | ✓ | ✓ |

combine and weight the extracted features. We follow this approach in our evaluations. LLM DI can be formalized as follows. First, after calculating over $n$ MIA scores with linear regression, an aggregated MIA score is obtained by $W \cdot \text{MIA}(x) = \sum_{i=1}^{n} w_i \text{MIA}_i(x)$. Here, $W = [w_1, ..., w_n]$ is the weight of the linear regressor, and $\text{MIA}(x)$ is a vector concatenating $n$ MIA scores. We label the suspect data as 0 and the held-out data as 1. Note that, $\text{MIA}(x)$ is calculated based on $f(x)$, but we omit $f$ for simplicity. Then, a hypothesis testing is conducted to verify if the held-out set has higher MIA score than the suspect set statistically. The null hypothesis can be formalized as follows.

$$\mathcal{H}_0 : \mathbb{E}_{\mathcal{D}_{\text{val}}}[W \cdot \text{MIA}(x_{\text{val}})] \leq \mathbb{E}_{\mathcal{D}_{\text{sus}}}[W \cdot \text{MIA}(x_{\text{sus}})]. \quad (1)$$

If the suspect set is part of the training set of $f$, the null hypothesis is rejected.

## 3. Failure Cases of DI

In this section, we dive deeper into the difficulties that arise from DI's assumption on the availability of an additional in-distribution held-out dataset. More precisely, we show that this assumption is extremely hard to meet in practice, even in the simplest setups—limiting the applicability of standard DI. Therefore, we collect blog posts written by a *single author* on topics from the *same domain* and split them randomly into a training and held-out set. We finetune an LLM on the training set, perform DI (Maini et al., 2024), and find that the method returns false positives, *i.e.,* it illegitimately claims that the model was trained on blog posts that it actually was not trained on (see Table 1). Our analysis highlights that despite the texts' homogeneity, there is a small distributional shift between the suspect and held-out sets that is not even easily distinguishable by Blind Baselines (Das et al., 2024), which causes DI to fail. This highlights the need to generate synthetic held-out data to benefit from DI in real-world copyright claims. We provide more details below and discuss its implications.

### 3.1. DI on a Single Author's Data

We consider a practical application of DI in copyright protection as detailed in Figure 1. In this scenario, an author has some published texts on the internet of which

they believe that they were illegitimately used by an LLM provider to train their model. The author provides these published works to an arbitrator, as a suspect set and some non-published blog-posts as held-out set from the same distribution, *i.e.,* with the same style, topics, etc. Then, the arbitrator performs DI to resolve the copyright claims.

To evaluate this setup in practice, we collect blog posts of a public blogger. The blogs are split into member, non-member, and held-out sets. To avoid any potential temporal or topic distributional shifts, we randomly shuffle all the collected blogs before splitting. In lack of the computational capacities to train an LLM from scratch, we finetune a Pythia model (Biderman et al., 2023) on the member set. The Pythia model is trained on the Pile dataset (Gao et al., 2020), so we only used blogs after the release date of the Pile to ensure that none of the blogs is part of the pretraining data. Also, we only finetune the target model on the member set for one epoch. This is to evaluate the performance of DI and our method in the most strict scenario, as Duan et al. (2024) show that MIAs perform better with more training epochs. Finally, we run DI. More detailed experiment configurations can be found in Section 5.1.

### 3.2. Metrics of Distributional Gap

Before analyzing the results, we introduce the metrics we use to quantify the distributional shift between the suspect and held-out sets. Following the approach of Blind Baselines (Das et al., 2024), we formulate the measurement of the distribution gap between two text datasets as a classification problem. In particular, the suspect set $\mathcal{D}_{\text{sus}}$ is randomly split into a classifier training split $\mathcal{D}_{\text{sus}}^{\text{train}}$ and a test split $\mathcal{D}_{\text{sus}}^{\text{test}}$. The held-out set $\mathcal{D}_{\text{val}}$ is also split into $\mathcal{D}_{\text{val}}^{\text{train}}$ and $\mathcal{D}_{\text{val}}^{\text{test}}$ in the same vein. Then, a classifier $g$ is optimized to distinguish the training splits $\mathcal{D}_{\text{sus}}^{\text{train}}$ and $\mathcal{D}_{\text{val}}^{\text{train}}$. Finally, we calculate the area under the curve (AUC) score of the classifier on the test splits $\mathcal{D}_{\text{sus}}^{\text{test}}$ and $\mathcal{D}_{\text{val}}^{\text{test}}$, which is used to measure the distributional gap between $\mathcal{D}_{\text{sus}}$ and $\mathcal{D}_{\text{val}}$.

The design of the classifier decides how the texts are vectorized and if the discrepancies between texts can be sufficiently captured. Das et al. (2024) apply a bag-of-words (BoW) classifier, which can only detect the differences in terms of word frequency. Instead, we build a GPT2-based classifier with two transformer blocks to also find the differences in grammar, content, styles, etc. between two text distributions. We train the classifier from scratch to avoid the impact of any pre-training data. Using only two transformer blocks of the GPT2 architecture avoids overfitting.

### 3.3. False Positive of DI

The AUC scores of the GPT2-based classifier in Table 1 show that there is a non-negligible distributional shift between the non-member and the held-out sets. The intuition

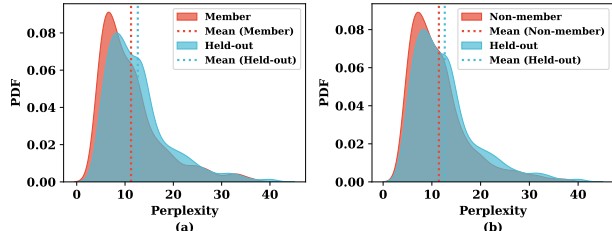

*Figure 2.* **Probability Distribution Function (PDF) of target model perplexities.** We show the comparison between (a) the member and held-out, and (b) the non-member and held-out sets.

behind this observation is that each blog has different content and topics, which brings different words across the non-member and held-out documents. The gap is enlarged when we sample more sequences from each blog post. Consequently, this distributional shift in texts also lead to a shift in the MIA score. As presented in Figure 2, the distributional shift in perplexities exists not only between member and held-out sets, but also between *non-member and held-out sets*. This shows that the inherent distributional shift among documents is entangled with the shift caused by membership signals in the MIA score, and makes DI fail to determine membership by simply detecting any distributional shift in the MIA score. This observation aligns with the p-values and predictions in Table 1, where we find that even this small distributional shift causes significant false positive rates during DI. This means that the DI falsely accuses the LLM provider of violating the copyright of an author. What is more is that this shortcoming of DI can be maliciously exploited: authors could deliberately provide held-out data from a different distribution than their suspect data to mislead DI and *illegitimately* accuse the LLM provider. As a solution to this problem, in the next section, we propose our approach on generating an adequate in-distribution held-out dataset synthetically.

## 4. Synthesizing Held-out Data

Our approach consists of two subsequent steps. First, we generate high-quality held-out data, then, we perform a calibration to account for the distribution shift that such generation can introduce.

### 4.1. Held-out Data Generation

We explore three approaches that leverage LLMs for generating held-out data based on provided suspect data with minimal distribution shift.

**In-context Learning.** As a naïve approach, we use GPT-4 models to paraphrase the suspect set with in-context learning (ICL) and evaluate the distributional shift between the original suspect texts and the paraphrased texts. Specifically, each prompt includes a few data points as demonstrations (shots) and requests the model to produce paraphrases for

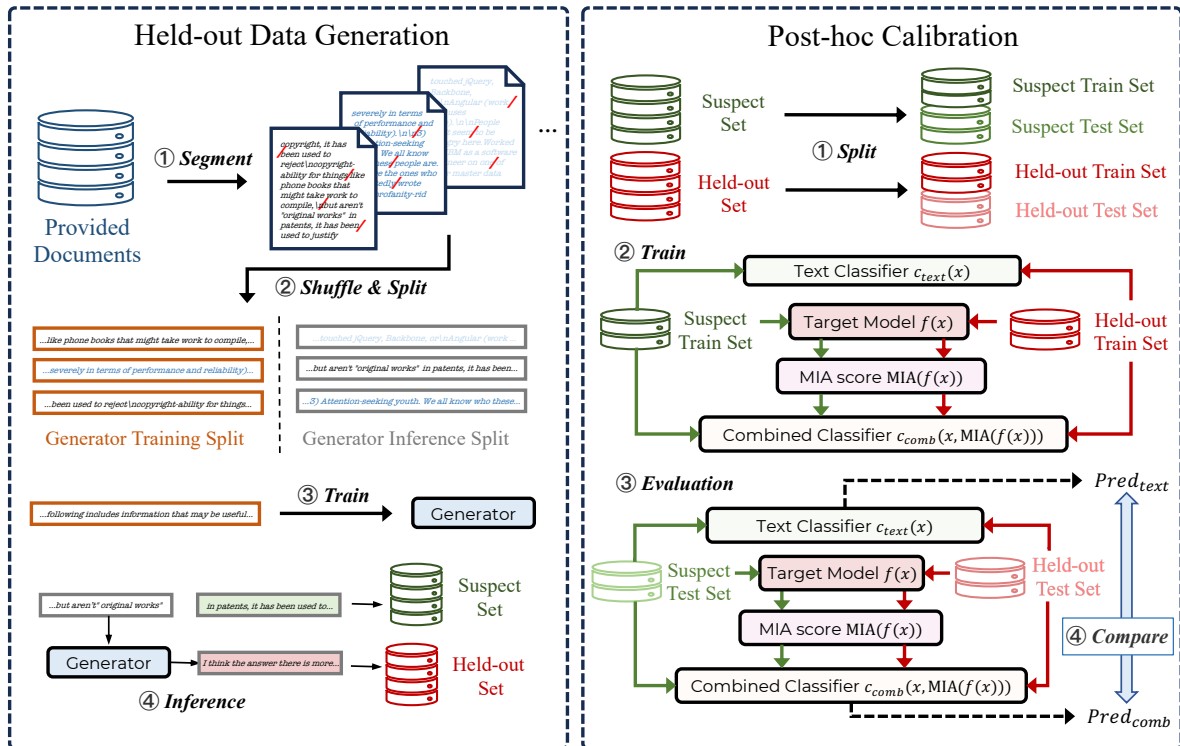

*Figure 3.* **Held-out Data Generation (Left Panel): (1)** The suspect dataset is first segmented into text snippets. **(2)** These snippets are shuffled and split into a generator training set and an inference set. **(3)** A generator model is trained on the suspect dataset using a suffix completion task. **(4)** The trained generator produces synthetic held-out data that mimics the suspect set. **Post-hoc Calibration (Right Panel): (1)** The suspect set and synthetic held-out set are split into training and test subsets. **(2)** A **text classifier** is trained to differentiate real from synthetic text. A **combined classifier** integrates textual features with DI membership signals from the target LLM. **(3)** The two classifiers are evaluated: the combined classifier should only outperform the text classifier if the suspect dataset was used for training. **(4)** A statistical comparison ensures that any signal detected is due to actual membership rather than distributional shifts.

another data point in the suspect set. We test two types of prompts: 1) ***Paraphrasing***, where we directly prompt the model to paraphrase a chosen data point, and 2) ***Text Completion***, where we prompt the model to complete a truncated data point. For text completion, the original suffix is used as the suspect data, and the generated suffix as the held-out data. Our results in Table 2 show that there is a significant distribution shift between the original and paraphrased samples for both types of prompts. Even a BoW classifier obtains a significant AUC of 76.2% when distinguishing between the original suspect vs paraphrased text and the GPT2 classifier can achieve 99.0% AUC. The reason is that there are many words (such as "remarkable" and "moreover") that appear much more frequently in the synthetic text than in the human-written text. Please refer to Appendix A for more detailed explanation of GPT-4-based generation and examples of generated texts.

**Preference Optimization.** We also adapt preference optimization methods (Rafailov et al., 2024; Xu et al., 2024) to the task of held-out data generation by changing from human preference to natural text preference. A more detailed explanation is presented in Appendix B. The AUC of the

*Table 2.* **Distributional shifts between the suspect set and synthetic held-out set.** The shifts are measured by Bag-of-Word (BoW) and GPT2 classifiers.

| Generation Method | BoW AUC (%) | GPT2 AUC (%) |
|---|---|---|
| ICL Paraphrasing | 76.2 | 99.0 |
| ICL Text Completion | 79.2 | 99.2 |
| Preference Optimization | 50.2 | 58.9 |
| Suffix Completion | **50.0** | **52.2** |

BoW classifier is similar to random guessing, which means frequent words can be greatly reduced in generated texts by this approach. However, the GPT2 classifier can still obtain an AUC of 58.9%. This shows that preference optimization still leaves distinguishable generation patterns that could be easily captured by a transformer-based classifier, limiting the data's usefulness as in-distribution held-out set.

**Suffix Completion.** The failure of the above methods demonstrates the difficulty of producing high-quality held-out data with a small enough distributional gap to the suspect data. To solve this problem, we design a generator training scheme that enables the generator to derive a suspect set from the author's provided documents, together with a held-

out set from the same distribution as this suspect set. As shown in Figure 3, we ① first segment the provided documents into multiple short sequences. ② All the sequences are shuffled and randomly split into a generator training split and a generator inference split. Then, ③ a low-rank adaption (LoRA) generator is finetuned on the training split with the cross-entropy loss for next-token prediction. Finally, ④ we segment each sequence in the generator inference split into two parts, and the generator predicts a synthetic suffix based on the prefix. Here, the original suffixes are used as the suspect set, and the synthetic suffixes as the held-out set. Note that, the training and inference sets are split on the shuffled text sequences rather than on the documents. This is to ensure that the text snippets from the generator training and inference splits are from the same distribution, such that the generator can achieve better generalization from the training to the inference set. Furthermore, we design a suffix completion task for generator inference. In this task, both the original suffix and the synthetic suffix share a common prefix. This approach ensures that the synthetic text maintains the same position within a sentence as its original counterpart, making the two suffixes directly comparable. Another important insight is that the generator can produce suffixes of higher quality when the sequence length is relatively short. Therefore, we limit the length of the sequences to no longer than 64 tokens for a smaller distributional gap. The results in Table 2 show that our method achieves a significantly small distributional shift, and even GPT2-based classifier can only achieve an AUC as low as 52.2%. For examples of our generative approach, please refer to Appendix D.2.

## 4.2. Post-hoc Calibration

Since the generation itself can introduce a distributional shift (natural vs generated) data, DI might yield false positives. This is because it would detect differences between suspect and held-out data also when they only differ in terms of distribution but not necessarily in membership. Therefore, we need to identify and mitigate this distribution shift.

To do so, we rely on an important observation: the generation shift between natural and synthesized data occurs in the textual space, while the shift caused by the potential membership of the suspect set exists in the target LLM's output space. This allows us to disentangle the two signals. By relying on our GPT-based **text-classifier** from Section 3.2, we can quantify the textual distribution shift caused by the generation. We denote this classifier by $c_{\text{text}}(x)$, where $x$ is the text input for which the classifier should decide if it is original or generated data. Inspired by Kazmi et al. (2024), we also define a second **MIA-classifier** with input signals from both the texts and the outputs of the target model, such that we can quantify the combined effects of generation and the membership signal. Concretely, we train a combined classifier $c_{\text{comb}}(x, \text{MIA}(f(x)))$ with inputs from both text

$x$ and the MIA signal $\text{MIA}(x)$ based on the outputs of $f$. Here, $\text{MIA}(x)$ can also be a vector by concatenating multiple MIA scores. We split both the suspect set and held-out sets into training and test splits. The two classifiers are optimized on the suspect train split $\mathcal{D}_{\text{sus}}^{\text{train}}$ and the held-out train split $\mathcal{D}_{\text{val}}^{\text{train}}$, and evaluated on the suspect test split $\mathcal{D}_{\text{sus}}^{\text{test}}$ and the held-out test split $\mathcal{D}_{\text{val}}^{\text{test}}$. By comparing the distributional shifts quantified by the MIA classifier and the shifts identified by the text classifier, we can separate the membership signals from the distribution gap caused by generation.

We design a hypothesis test to statistically verify if the combined classifier quantifies a larger distributional shift between the suspect and held-out data than the text classifier, namely the *difference comparison t-test*. The t-test is conducted on the test splits $\mathcal{D}_{\text{sus}}^{\text{test}}$ and $\mathcal{D}_{\text{val}}^{\text{test}}$, but we abbreviate them as $\mathcal{D}_{\text{sus}}$ and $\mathcal{D}_{\text{val}}$ for simplicity. During the t-test, we first sample a suspect data point $x_{\text{sus}} \in \mathcal{D}_{\text{sus}}$ and pair it with its corresponding generated counterpart $x_{\text{val}} \in \mathcal{D}_{\text{val}}$. Note that, $x_{\text{sus}}$ and $x_{\text{val}}$ are original and generated suffixes, which are both continuations of a common prefix. For every such original/held-out pair, we quantify the shift caused by *generation* with the text classifier as $c_{\text{text}}(x_{\text{val}}) - c_{\text{text}}(x_{\text{sus}})$. We also quantify the combined effects caused by *generation and membership signal* with the combined classifier as $c_{\text{comb}}(x_{\text{val}}) - c_{\text{comb}}(x_{\text{sus}})$. In particular, $x_{\text{val}}$ and $x_{\text{sus}}$ are labeled as 1 and 0, respectively. $c(x)$ denotes the predicted probability of classifier $c$ on data point $x$, which ranges between 0 and 1. If the membership signal is present, the combined effects will be stronger than the generation effect alone, and the predicted probability will be slightly more accurate for the combined classifier, i.e. $c_{\text{comb}}(x_{\text{val}}) - c_{\text{comb}}(x_{\text{sus}}) > c_{\text{text}}(x_{\text{val}}) - c_{\text{text}}(x_{\text{sus}})$. To this end, we formalize the following null hypothesis for our t-test:

$$\mathcal{H}_0 : \mathbb{E}_{x_{\text{val}} \in \mathcal{D}_{\text{val}}, x_{\text{sus}} \in \mathcal{D}_{\text{sus}}}[c_{\text{comb}}(x_{\text{val}}) - c_{\text{comb}}(x_{\text{sus}})] \leq \\ \mathbb{E}_{x_{\text{val}} \in \mathcal{D}_{\text{val}}, x_{\text{sus}} \in \mathcal{D}_{\text{sus}}}[c_{\text{text}}(x_{\text{val}}) - c_{\text{text}}(x_{\text{sus}})]. \quad (2)$$

The difference comparison t-test is performed multiple times with different random seeds, and the p-values are aggregated with Sidac correction (Šidák, 1967).

By introducing a dual-classifier approach along with a statistical test, we can statistically distinguish distributional shifts caused by actual membership signals from those caused by generation. Further results in Section 5.4 show that this approach can prevent false positives in dataset inference effectively.

## 4.3. Weight Constraint

In this section, we explain why and how we apply a weight constraint when computing the importance of different MIA scores. In the original DI, the aggregated MIA score is compared between the held-out and the suspect sets. We define the difference in aggregated MIA score between the

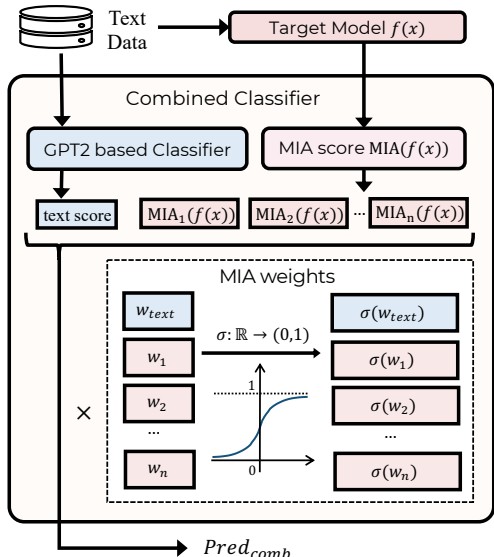

*Figure 4.* **Weight Constraint.** The weights for the MIA and text scores are constrained to (0,1) with Sigmoid function.

two sets $y_{\text{diff}}$ as follows:

$$
\begin{aligned}
y_{\text{diff}} &= \mathbb{E}[\sum_{i=1}^{n} w_i \text{MIA}_i(x_{\text{val}})] - \mathbb{E}[\sum_{i=1}^{n} w_i \text{MIA}_i(x_{\text{sus}})] \\
&= \sum_{i=1}^{n} w_i (\mathbb{E}[\text{MIA}_i(x_{\text{val}})] - \mathbb{E}[\text{MIA}_i(x_{\text{sus}})]) \\
&> 0 \text{ if } \mathcal{D}_{\text{sus}} \text{ is member set, otherwise} \leq 0.
\end{aligned}
\tag{3}
$$

Here, $w_i \in \mathbb{R}$ is the weight for the MIA score $\text{MIA}_i$. Assuming that $\mathcal{D}_{\text{sus}}$ and $\mathcal{D}_{\text{val}}$ are i.i.d., we have $\mathbb{E}[\text{MIA}_i(x_{\text{val}})] > \mathbb{E}[\text{MIA}_i(x_{\text{sus}})]$ on member set and $\mathbb{E}[\text{MIA}_i(x_{\text{val}})] \approx \mathbb{E}[\text{MIA}_i(x_{\text{sus}})]$ on non-member set for each MIA score. Therefore, we have $y_{\text{diff}}$ close to 0 on the non-member set, regardless of the weights $w_i$. However, when we synthesize the held-out set with a generator, there can be a small distributional shift between $\mathcal{D}_{\text{sus}}$ and $\mathcal{D}_{\text{val}}$. With this shift, we can have $\mathbb{E}[\text{MIA}_i(x_{\text{val}})] < \mathbb{E}[\text{MIA}_i(x_{\text{sus}})]$ on non-member set. This is often the case for generated text, because the generator usually produces held-out texts that are *simpler* than human-written texts, therefore causing the generated held-out texts to have smaller perplexity. This affects most perplexity-based methods, such as LOSS, Min-K%, and Zlib ratio. Consequently, the linear regression algorithm can assign negative weight $w_i$ to such MIA scores, which causes $y_{\text{diff}} > 0$ on the non-member set and therefore high false positive rates. To ensure that this generation shift does not add up to a falsely high $y_{\text{diff}}$, we constrain the weights to be positive. As shown in Figure 4, the weights are projected from $\mathbb{R}$ to $(0, 1)$ with Sigmoid function $\sigma(x) = \frac{1}{1+e^{-x}}$. With such a weight constraint, the linear regression only assigns a small weight $w_i$ for $\text{MIA}_i$ if $\mathbb{E}[\text{MIA}_i(x_{\text{val}})] < \mathbb{E}[\text{MIA}_i(x_{\text{sus}})]$, avoiding false positives in many cases. We also present the

*Table 3.* **Results for single author blog posts.** Here, p-value $<$ 0.05 indicates the suspect set is member set.

| True Membership | AUC$_{\text{Text}}$ (%) | AUC$_{\text{Comb}}$ (%) | P-value | Inferred Membership |
|:---:|:---:|:---:|:---:|:---:|
| ✓ | 53.8 | 55.6 | 0.01 | ✓ |
| ✗ | 53.8 | 53.9 | 0.13 | ✗ |

empirical analysis of the weight constraint in Section 5.4 and an example for the weight constraint in Appendix G.

## 5. Experimental Evaluation

We start by introducing our experimental setup, further detailed in Appendix D. Then, we present the results of DI executed based on our generated held-out data. We also perform ablation studies to investigate the contribution of each component in our proposed method. Finally, we analyze the impact of t-test sample size and the classifier architecture.

### 5.1. Experimental Setup

**Single author data.** We collect 1400 blog posts from a single author. All figures, tables, videos, and hyperlinks are removed during pre-processing and only plain text is used for evaluation. We sample 450 posts as member data and finetune a Pythia 410M deduplicated model as target model. The other posts are held out as non-member and held-out sets for the evaluation.

**More Complicated Dataset and Model.** We also evaluate our method on the Pile dataset (Gao et al., 2020), which is much more complicated and has subsets of diverse types of texts. We use the de-duplicated version of Pythia 1B model as the target model. The training split of the Pile dataset is used as member data, and the held-out and test split is used as non-member data. Here, we only evaluate Pile subsets that are free from copyright issues. Please also refer to Appendix C for detailed configuration on the Pile.

**Implementation Details** We finetune a Llama 3 8B model (Dubey et al., 2024) with LoRA as the generator. For both types of datasets, we split 2,000 sequences as the generator inference set, and the others as the generator training split. Both text classifier and combined classifier are trained on 1,000 synthetic held-out data and 1,000 suspect data for each dataset. Our proposed t-test is also conducted on 1,000 synthetic held-out data and 1,000 suspect data. More implementation details can be found in Appendix D. We also provide an analysis of hyperparameter sensitivity in Appendix K.

### 5.2. Results for Single Author Dataset

The experimental results on the single author dataset are presented in Table 3. On the member set, the combined classifier $c_{\text{comb}}$ outperforms the text classifier $c_{\text{text}}$, by a large mar-

*Table 4.* **Results for different Pile subsets.** *True* represents the true membership while *Inferred* denotes the inferred membership. Our generation is successful if these two align.

| Subset | True | $AUC_{Text}$ (%) | $AUC_{Comb}$ (%) | P-value | Inferred |
|---|---|---|---|---|---|
| Pile-CC | ✓ | 55.6 | 58.2 | 0.002 | ✓ |
| | ✗ | 55.3 | 53.3 | 0.99 | ✗ |
| Wikipedia | ✓ | 56.0 | 56.8 | 0.04 | ✓ |
| | ✗ | 54.9 | 52.4 | 1.00 | ✗ |
| ArXiv | ✓ | 53.6 | 59.1 | <0.001 | ✓ |
| | ✗ | 53.1 | 53.3 | 0.74 | ✗ |
| NIH ExPorter | ✓ | 56.8 | 57.7 | 0.02 | ✓ |
| | ✗ | 55.6 | 53.3 | 1.00 | ✗ |
| FreeLaw | ✓ | 52.8 | 58.4 | <0.001 | ✓ |
| | ✗ | 51.4 | 53.9 | 0.09 | ✗ |
| Ubuntu IRC | ✓ | 53.4 | 55.8 | 0.01 | ✓ |
| | ✗ | 53.4 | 54.9 | 0.33 | ✗ |
| PubMed Central | ✓ | 54.6 | 58.1 | <0.001 | ✓ |
| | ✗ | 54.7 | 55.5 | 0.11 | ✗ |
| Github | ✓ | 53.6 | 55.7 | 0.003 | ✓ |
| | ✗ | 53.9 | 55.4 | 0.07 | ✗ |
| EuroParl | ✓ | 51.0 | 57.0 | <0.001 | ✓ |
| | ✗ | 51.4 | 53.9 | 0.07 | ✗ |
| PhilPapers | ✓ | 55.1 | 59.1 | <0.001 | ✓ |
| | ✗ | 61.1 | 56.0 | 0.99 | ✗ |
| HackerNews | ✓ | 56.7 | 60.7 | <0.001 | ✓ |
| | ✗ | 58.0 | 56.7 | 0.43 | ✗ |
| Enron Emails | ✓ | 54.5 | 58.0 | <0.001 | ✓ |
| | ✗ | 54.6 | 52.9 | 0.99 | ✗ |
| StackExchange | ✓ | 54.3 | 60.1 | <0.001 | ✓ |
| | ✗ | 53.0 | 55.0 | 0.06 | ✗ |
| PubMed Abstracts | ✓ | 54.5 | 59.0 | <0.001 | ✓ |
| | ✗ | 53.8 | 53.4 | 0.90 | ✗ |
| USPTO Backgrounds | ✓ | 52.8 | 57.1 | 0.001 | ✓ |
| | ✗ | 52.8 | 52.2 | 0.97 | ✗ |
| DM Mathematics | ✓ | 53.9 | 55.5 | 0.002 | ✓ |
| | ✗ | 54.0 | 51.3 | 1.00 | ✗ |

*Table 5.* **Ablation studies of our approach.** *Setting 1-3:* replacing our generation method with baselines. *Setting 4-5:* removing key designs from our generation method. *Setting 6:* without post-hoc calibration. *Setting 7:* without weight constraint. *Setting 8:* our complete method.

| Setting | Configuration | True Membership | P-value | Inferred Membership |
|---|---|---|---|---|
| 1 | w/o Suffix Completion *(ICL Paraphrasing)* | ✓ | 1.0 | ✗ |
| | | ✗ | 1.0 | ✗ |
| 2 | w/o Suffix Completion *(ICL Text Completion)* | ✓ | 1.0 | ✗ |
| | | ✗ | 1.0 | ✗ |
| 3 | w/o Suffix Completion *(Preference Optimization)* | ✓ | 1.0 | ✗ |
| | | ✗ | 1.0 | ✗ |
| 4 | w/o Segment and Shuffle | ✓ | 1.0 | ✗ |
| | | ✗ | 1.0 | ✗ |
| 5 | w/o Suffix Comparison | ✓ | 1.0 | ✗ |
| | | ✗ | 1.0 | ✗ |
| 6 | w/o Post-hoc Calibration *(Original T-test in DI)* | ✓ | <0.001 | ✓ |
| | | ✗ | <0.001 | ✓ |
| 7 | w/o Weight Constraint | ✓ | 0.004 | ✓ |
| | | ✗ | 0.43 | ✗ |
| 8 | Ours | ✓ | <0.001 | ✓ |
| | | ✗ | 1.0 | ✗ |

text, academic writing, and code using our method for generating the held-out data. The results also show that our generation method generalizes well to documents with different lengths, ranging from 1 KB (Wikipedia) to 70 KB (PhilPapers). Moreover, our proposed method generalizes well to texts from different domains and languages, *e.g.,* medical (PubMed Central), legal (FreeLaw), and multilingual (EuroParl) domains. Notably, the p-values for our difference comparison t-test are significantly lower than 0.05 on all the evaluated member sets, and higher than 0.1 on all the non-member sets. Please refer to Appendix F and Appendix I for the results on different model sizes and model architectures.

### 5.4. Ablation on Post-hoc Dataset Inference

We conduct ablation studies to separately analyze the contribution of the three components in our held-out data generation: suffix completion, calibrating, and weight constraint.

**Suffix Completion.** As presented in Table 2, our proposed sequence completion scheme can synthesize held-out texts with a distribution much more similar to the suspect texts when compared with the baseline methods. In addition to the AUC results, we also show that the baseline generation methods cannot produce reliable held-out sets even when combined with our post-hoc calibration and weight constraint in Table 5. In particular, we replace our generation scheme with three baselines, including ICL paraphrasing, ICL text completion, and preference optimization. The p-values are presented as Setting 1-3. We also remove two key designs in our generation method, 1) Segment and Shuffle, and 2) Suffix Comparison, as shown in Setting 4-5. In all

gin of 1.8% AUC score. Moreover, the observed p-value of 0.01 strongly supports the alternative hypothesis, indicating that the superior performance of $c_{comb}$ over $c_{text}$ is statistically significant. This enables our method to correctly identify that the target set is part of the training set. For the non-member set, $c_{comb}$ and $c_{text}$ achieve comparable AUC scores, with a p-value of 0.13 that significantly exceeds the threshold of 0.05. This result confirms the ability of our approach to correctly identify non-member texts as such, thus avoiding the false positives that occur with the original LLM DI approach. Here, we finetune the target model on the single author dataset with LoRA for one epoch. We also present the results with other fine-tuning setups in Appendix H.

### 5.3. Results for Pile Datasets

The results of different Pile subsets are shown in Table 4. We observe that DI correctly predicts the membership of datasets from diverse domains and styles, including plain

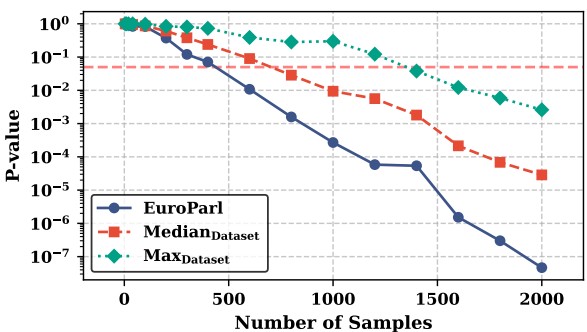

*Figure 5.* **The p-values of member sets with change in sample size.** Median$_\text{Dataset}$ denotes the median p-value of different datasets, and Mean$_\text{Dataset}$ is the maximum p-value of all subsets. Number of samples refers to the total size of both suspect and validation sets.

the above settings, the p-values for both member and non-member sets are 1.0, which indicates that the $c_\text{text}$ has better or similar performance when compared with $c_\text{comb}$. The reason behind the observation is that the distributional shift caused by the generation is much larger than the shift induced by the membership signal, such that $c_\text{comb}$ does not outperform $c_\text{text}$ even with extra membership inputs on the member set. Consequently, the DI predicts both sets as non-member and suffers from false negatives.

**Post-hoc Calibration.** We replace our calibration method with the original DI without calibration, as shown in Setting 6 of Table 5. Specifically, only a linear classifier is optimized to aggregate different MIA metrics and output the final prediction score. Furthermore, the t-test is conducted directly between the predictions on the target set and the ones on the held-out set. We observe that the p-values under this condition are extremely low for both member and non-member sets, and DI has false positive in this case. This observation aligns with results in Section 3, where we show that even a small distributional shift causes a significantly small p-value in the original DI. Therefore, our post-hoc calibration approach is crucial to evaluating the distributional shift caused only by membership signals.

**Weight constraint.** As explained in Section 4.3, the weight constraint avoids summing the distributional shift caused by generation to the final MIA prediction when the direction of the generation shift is different from that caused by the membership signal. As shown in Setting 7 of Table 5, applying the constraint leads to a much lower p-value on the member set and much higher on non-member set, which helps our method make a more accurate prediction about the membership.

### 5.5. Analysis of Sample Size

We also set out to analyze how the sample size in our proposed t-test affects the statistical confidence of DI with our generated held-out data. Here, the sample size is the total

*Table 6.* **The AUC of different classifier architectures.**

| Architecture | AUC$_\text{Text}$ | Training Time |
|---|---|---|
| all-MiniLM-L6-v2 | 50.8 | **0.3** |
| BERT | 51.2 | 2.0 |
| Llama3-8B, Pre-trained+LoRA | 53.2 | 65.1 |
| GPT2, Pre-trained+LoRA | 53.0 | 26.2 |
| GPT2, Pre-trained+Full Finetuned | 52.3 | 36.8 |
| GPT2, 2 Layers+Initialized | **53.3** | 0.5 |

number of the suspect and held-out set, which is also the number of queries made to the target model. The two sets are of the same size, as they are produced in a pairwise manner. We observe from Figure 5 that, as the number of samples increases, DI exhibits improved detection capability of training data. Notably, with fewer than 1,000 samples, DI achieves statistical significance ($p < 0.05$) across most of the evaluated datasets. When increasing the sample size to 2k queries, the method demonstrates even stronger statistical significance ($p < 0.01$) consistently across all datasets.

### 5.6. Choice of Classifier

We explore different text classifier architectures and present the results for different architectures and different parameter sizes in Table 6. The results show that the simple GPT2-based classifier with 2 layers and initialized weights can achieve the best AUC in our experimental settings. Additionally, this lightweight classifier has a significantly short training time, making the method more practical when faced with more queries. Therefore, we choose this 2-layer GPT2-based classifier as our text classifier. During our experiments, we consider the scenario where the author can only provide a limited number of tokens, so stronger text classifiers, such as Llama and full GPT2 models, can be easily overfitted. In real-world applications, an arbitrator is suggested to select the most suitable text classifier based on their specific conditions regarding data size, data type, and computation resources.

## 6. Conclusions

We propose how to *synthetically generate* an in-distribution held-out dataset to enable the real-world application of DI. Therefore, we solve two critical challenges, namely (1) creating high-quality, diverse synthetic data that accurately reflects the original distribution and (2) bridging likelihood gaps between real and synthetic data. Our solution relies on designing a data generator training scheme based on a suffix-based completion task and post-hoc calibration to align the likelihood gaps between real and synthetic data. Through extensive experimental evaluation, we highlight that our method enables a robust DI and correctly identifies training data while achieving a low false positive rate. This shows our method's reliability to support copyright owners to make legitimate claims on data usage for real-world litigations.

## Impact Statement

Crawling of data for training LLMs is becoming pervasive, with model training companies scraping vast spans of the internet in order to find high-quality data. Given the strong correlation between data quality and model performance, many content creators want to protect their work from being trained on, without their consent. Claiming that one's data has been trained on, only with access to the suspect LLM, has thus far stayed near impossible. Our work takes a leap forward by allowing content creators to 'post-hoc' infer if their data on the internet was trained on by leveraging synthetic data. This means that authors, bloggers, and columnists with decades of internet data can now leverage our method in order to claim their rightful ownership. Our work aims to serve as an important tool in future copyright litigations, in particular in their 'discovery' period.

## Acknowledgment

This work was supported by the German Research Foundation (DFG) within the framework of the Weave Programme under the project titled "Protecting Creativity: On the Way to Safe Generative Models" with number 545047250. Responsibility for the content of this publication lies with the authors.

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

# A. Details of ICL Generation

We experiment with GPT-4-Turbo and prompt it using in-context-learning learning to paraphrase the suspect data. Each prompt includes a few data points as demonstrations (shots) and requests the model to produce paraphrases for the suspect set. We test two types of templates to prompt GPT-4-Turbo model with in-context-learning (ICL). One is to paraphrase the given sample based on the examples, and the other is to complete the text with part of the sentence.

## A.1. Text Completion Prompt

In the text completion prompt, we give ten examples from the author provided documents, and prompt the GPT-4-Turbo model to complete the given sample, which is shown at the end of the prompt. Here is an example of this kind of queries:

*Input:*

```
I have some text samples. Please help me complete the last sample based on my
    example samples.
A device may be provided to sense an environment's present temperature. For
    example, [...] a thermistor that is robust and reliable and that can be
    fabricated at a reasonable cost.
[...]
This invention relates to a method of synthesizing musical tones.A musical tone
    or natural tone produced by an acoustic or natural musical instrument can
    [...]  With the music synthesizer it is considerably difficult to obtain
    dynamic audio spectra.pneumoniae.
Radio communications receivers commonly receive and decode messages which contain
     information. A part of this information may be [...]a single radio will
    transmit messages to the remainder of the group. These radio
```

*Output:*

```
messages are selectively received by the grouped radios which decode the identity
     information within the [...] and secure group communications in diverse and
    often challenging operational environments.
```

## A.2. Paraphrasing Prompt

In the paraphrasing prompt, we give ten examples from the author provided documents, and prompt the GPT-4-Turbo model to paraphrase the last. Here is an example of this kind of queries:

*Input:*

```
Paraphrase the last sample according to the other samples.
1. Field of the Invention This invention relates generally to fluid injectors.
    More particularly, this invention relates to a proportional needle control
    fuel injector having a [...] The needle valve is moveable between a first
    position closing the injection fuel outlet port and a second position to open
     to injection fuel outlet.
[...]
The present invention relates to the structure of a casing of a miniature
    portable [...] is forcibly urged away from the casing by a tool, it is likely
     that the casing is scratched or the base plate is deformed.
This invention relates to fuel cells and, in particular, to improvements in the
    catalyst used in internal  [...]  fuel cell in which the catalyst assembly
    has minimum displacement and permits vertical and horizontal orientations of
    the fuel cell.
```

*Output:*

```
This invention pertains to fuel cells, specifically addressing enhancements in
    catalysts utilized  [...] and efficiency of internal reforming fuel cells by
    mitigating previous limitations and improving overall system functionality.
```

### A.3. Distribution Shift Analysis

The distributional shifts are quantified in BoW AUC and GPT2 AUC as shown in Table A1. As observed, both types of prompt templates produces synthetic texts with large distributional shifts to the suspect sets. Notably, the GPT2-based classfier can achieve as much as an AUC of 99.2%.

*Table A1.* **Distributional shifts between the suspect set and GPT-4-Turbo generated validation set.**

| Template Type | BoW AUC(%) | GPT2 AUC(%) |
|---|---|---|
| ICL Text Completion | 79.2 | 99.2 |
| ICL Paraphrasing | 76.2 | 99.0 |

## B. Details of Preference Optimization Generation

Preference optimization methods focus on optimizing a pre-trained LLM based on human preference (Rafailov et al., 2024; Xu et al., 2024). Particularly, LLMs iteratively produce random generations, then human annotators are requested to label the generations as chosen or rejected, and the LLMs are further optimized according to this human feedback. We note that, we can leverage preference optimization approaches to make our generator model prefer the human-written texts over synthetic data, thus producing texts with a more similar distribution to natural texts. Here, we instantiate the preference optimization scheme with a state-of-the-art method, the simple preference optimization (SimPO) (Meng et al., 2024). During each training iteration, the human-written suspect data are always labeled as chosen and the generations from the last iteration are marked as rejected. As noted in Section 4.1, this approach improves significantly upon prompted paraphrasing, but still causes a large distributional shift between the suspect set and the generated held-out set.

*Table A2.* **Segmentation configurations for different Pile subsets.**

| Subset | Number of Test Set in Pile | Chosen Split Size | Max. Snippets per Document | Number of Tokens per Snippet |
|---|---|---|---|---|
| Pile-CC | >3000 | 3000 | 30 | 64 |
| StackExchange | >4000 | 4000 | 5 | 64 |
| PubMed Abstracts | >6000 | 6000 | 5 | 64 |
| Wikipedia (en) | >3000 | 3000 | 30 | 64 |
| USPTO Backgrounds | >6000 | 6000 | 10 | 32 |
| PubMed Central | >500 | 500 | 100 | 64 |
| FreeLaw | >4000 | 4000 | 5 | 32 |
| ArXiv | >200 | 200 | 100 | 32 |
| NIH ExPorter | >4000 | 4000 | 5 | 64 |
| HackerNews | >3000 | 3000 | 10 | 64 |
| Github | >2000 | 2000 | 20 | 64 |
| Enron Emails | 1957 | 1957 | 50 | 32 |
| DM Mathematics | >1500 | 1500 | 50 | 32 |
| EuroParl | 290 | 290 | 200 | 32 |
| PhilPapers | 132 | 132 | 100 | 32 |
| Ubuntu IRC | 43 | 20 | 2000 | 32 |

## C. Pile Dataset Segmentation

We present the details for the configurations of Pile subset in Table A2. We note that, it is claimed that the following Pile subsets may have copyright issues and cannot be included for evaluation: Books3, OpenWebText2, Gutenberg (PG-19), OpenSubtitles. BookCorpus2, and YoutubeSubtitles. For most subset there are documents that are much longer than the other documents, which causes that too many snippets are sample from these documents if all snippets are used. Therefore, we set a maximum snippet for each document on each subset according to the median lengths of the documents. Also, we note that our approach can achieve good performance on most subsets with only 32 tokens. For certain subsets, we use

a token length of 64 for a stronger membership signal. The average token number for the Pile subset is 45. In practical applications, we suggest choosing the token numbers when the text-classifier has the minimal AUC to avoid distributional shift as much as possible. As a more conservative approach, this avoids making false accusations of copyright violation.

## D. Implementation Details

### D.1. Generator

The LoRA rank for the generator is 32. The generator is trained for 100 epochs, and the learning rate is set to $2 \times 10^{-4}$. We set a warm-up ratio of 0.03, and a linear scheduler is used to dynamically adjust the learning rate.

### D.2. Text and Combined Classifiers

For both the text and the combined classifier, we leverage the basic architecture of the GPT2 classifier with an extra linear layer. Specifically, the classifier has only two layers, with an embedding dimension of 1600 and an attention head number of 25. As explained in Section 4.3,, we apply a weight constraint to the linear layer. The GPT2-based classifier is optimized for 20 epochs, and the linear layer is further optimized for 200 epochs.

## E. Examples of Synthetic Texts

In this section, we provide some examples of the synthetic texts on the Pile dataset. Here, prefix denotes the first half of the generated text, real suffix refers to the original suffix of the natural text, and generated suffix refers to the synthetic completion based on the prefix. We observe that, the generated suffixes are reasonable continuation of the prefixes. The generated suffixes also align with the style of each dataset and do not overfit to the content of the real suffixes.

### E.1. Pile-CC

*Prefix:*

```
are excited about and also what we hoped to see from this years E3!
```

*Real suffix:*

```
From the surprising new Spider-Man PS4 game to the bizarre We Happy Few and
```

*Generated suffix:*

```
Let us know your thoughts on this monologue as we are preparing for our next
```

### E.2. StackExchange

*Prefix:*

```
var FKEntityListWithCastCopy = new debiteur().GetType().GetProperty(\""
```

*Real suffix:*

```
schakeling\").GetValue(dbEntry) as List<FKEntity>;//Just
```

*Generated suffix:*

```
FKEntityList\").GetValue(instance, null);\n          foreach(var t in FKEntity
```

### E.3. PubMed Abstracts

*Prefix:*

```
were calculated using the Kaplan-Meier method. Of the 117 patients in
```

*Real suffix:*

```
whom data were analyzed, 103 had follow-up MR or CT images and 14 patients were
```

*Generated suffix:*

```
the study (76 with UC and 41 with DC), 45 patients required proctocolic resection
```

### E.4. Wikipedia (en)

*Prefix:*

```
Em is going away for a while. While it's not up to the standard
```

*Real suffix:*

```
of "Mockingbird," it is more fully realized than the two other new
```

*Generated suffix:*

```
of their three previous albums, cattle call is still an enjoyable romp,
```

### E.5. USPTO Backgrounds

*Prefix:*

```
1. Field of the Invention\nThis invention relates to a storage device for
   athletic equipment and, in particular, to a portable storage device for
   transporting and retaining
```

*Real suffix:*

```
elongate items of athletic equipment such as hockey sticks and related athletic
   equipment.\n2. Discussion of Related Art\nNumerous team athletic activities
   require individual players on the
```

*Generated suffix:*

```
multiple pairs of basketballs.\n2. Description of the Related Art\nDuring the
   summer and other periods when there is an extended break from an athletic
   school or program
```

### E.6. PubMed Central

*Prefix:*

```
example, both cycles apply Lewis acidic metal centers to bind the monomers (ep
```

*Real suffix:*

```
oxide or lactone), and both invoke labile metal alkoxide intermediates as
```

*Generated suffix:*

```
oxides or cyclic carbonates), but the axes of the metallacycle in
```

### E.7. FreeLaw

*Prefix:*

```
Court, 638 P.2d 65 (Colo.1981
```

*Real suffix:*

```
Here, the juvenile court denied the GAL's motions because it did not want
```

*Generated suffix:*

```
), cert. denied, 454 U.S. 1146, 102
```

### E.8. Arxiv

*Prefix:*

```
up and vice versa. In contrast, fundamentalists expect the price to track its
```

*Real suffix:*

```
fundamental value. Orders from this type of agent may be written as\n\n$$D
```

*Generated suffix:*

```
underlying fundamentals up and down, but given sufficient acceleration the price
    might \u201crun away
```

### E.9. NIH ExPorter

*Prefix:*

```
attachment and growth, respectively. Together with an industrial sponsor, Vaxiron
    ,
```

*Real suffix:*

```
Inc., we will develop quality control tools and metrics for assessing vaccine
    antigen formulations,
```

*Generated suffix:*

```
the applicant has carried out clinical trials of different vaccine candidates
    based on different viruses for
```

### E.10. Github

*Prefix:*

```
.string \"reach only by using a BIKE technique.$\"\n\nRoute110_Text_
```

*Real suffix:*

```
16EEF6:: @ 816EEF6\n\t.string \"Which
```

*Generated suffix:*

```
16F381:: @ 816F381\n\t.string \"ROUTE {ROAD
```

**E.11. Enron Emails**

*Prefix:*

Lay.  He went on to say that Kenneth was Dewayne Re

*Real suffix:*

es' cousin  and started telling about all of your fine attributes and what a

*Generated suffix:*

ams' direct \nreport and that it would be extremely difficult for Kenneth to get

**E.12. EuroParl**

*Prefix:*

het mondeling amendement op schrift heeft gekregen.\nIk st

*Real suffix:*

el voor om niet te spreken over \"de Raad en de lidstat

*Generated suffix:*

akk voor de uitnodiging om tijdens uw volgende bij

**E.13. PhilPapers**

*Prefix:*

distribute well among [the gods who fought with him] their titles and privileges

*Real suffix:*

" (885, cf. 66\u201367 and 74); to swallow

*Generated suffix:*

 (17.1). Orderly distribution of praise for the victory is re

**E.14. Ubuntu IRC**

*Prefix:*

about setting up reoccuring status meetings?\n<dfarning> should we start

*Real suffix:*

holding those or is it too soon?\n<dfarning> Luke will be joining

*Generated suffix:*

with a status meeting or a design meeting?\n<manusheel> dfarning

### E.15. HackerNews

*Prefix:*

```
Angular (work just uses Dojo).\n\nPeople don't seem to
```

*Real suffix:*

```
be hungry here.\n\n------\nlewispollard\nWorked for IBM as a software engineer on
    one of
```

*Generated suffix:*

```
care that it's adding yet another ~20KB per page. We're\nsaying no
```

## F. Generalization to Different Model Sizes

Here, we evaluate the performance of our method on three different model sizes of the Pythia model: 1.4B, 2.8B, and 6.9B. We use an outlier removal ratio $r = 0.01$ for Pythia-6.9B, $r = 0.05$ for Pythia-2.8B, $r = 0.1$ for Pythia-1.4B, $r = 0.15$ for Pythia-1B, $r = 0.2$ for Pythia-410M. All the tested models are the deduplicated versions. The results demonstrate that our proposed method generalizes well across different model sizes.

*Table A3.* **Results for different sizes of Pythia Models.** *True* represents the true membership while *Inferred* denotes the inferred membership. Our generation is successful if these two align.

| Subset | True | Pythia-410M | | | | Pythia-1.4B | | | | Pythia-2.8B | | | | Pythia-6.9B | | | |
|---|---|---|---|---|---|---|---|---|---|---|---|---|---|---|---|---|---|
| | | $AUC_{Text}$ (%) | $AUC_{Comb}$ (%) | P-value | Inferred | $AUC_{Text}$ (%) | $AUC_{Comb}$ (%) | P-value | Inferred | $AUC_{Text}$ (%) | $AUC_{Comb}$ (%) | P-value | Inferred | $AUC_{Text}$ (%) | $AUC_{Comb}$ (%) | P-value | Inferred |
| Pile-CC | ✓ | 54.9 | 57.4 | 0.009 | ✓ | 55.5 | 57.5 | 0.007 | ✓ | 56.2 | 57.8 | 0.02 | ✓ | 55.9 | 58.2 | 0.006 | ✓ |
| | ✗ | 54.2 | 51.0 | 1.00 | ✗ | 55.1 | 53.9 | 0.89 | ✗ | 54.0 | 52.8 | 0.94 | ✗ | 54.8 | 55.1 | 0.30 | ✗ |
| ArXiv | ✓ | 54.3 | 60.5 | <0.001 | ✓ | 54.0 | 59.3 | <0.001 | ✓ | 53.8 | 57.7 | <0.001 | ✓ | 53.7 | 57.0 | <0.001 | ✓ |
| | ✗ | 52.8 | 52.9 | 0.87 | ✗ | 53.8 | 54.0 | 0.64 | ✗ | 52.3 | 53.1 | 0.67 | ✗ | 52.4 | 53.6 | 0.39 | ✗ |
| FreeLaw | ✓ | 52.9 | 58.3 | <0.001 | ✓ | 52.6 | 57.3 | <0.001 | ✓ | 52.2 | 57.4 | <0.001 | ✓ | 52.2 | 56.7 | <0.001 | ✓ |
| | ✗ | 52.3 | 53.6 | 0.41 | ✗ | 52.1 | 54.7 | 0.07 | ✗ | 52.2 | 54.6 | 0.05 | ✗ | 51.9 | 54.6 | 0.05 | ✗ |
| PubMed Central | ✓ | 54.7 | 58.0 | <0.001 | ✓ | 54.4 | 58.4 | <0.001 | ✓ | 55.1 | 57.9 | 0.002 | ✓ | 54.7 | 57.2 | 0.002 | ✓ |
| | ✗ | 55.2 | 55.4 | 0.24 | ✗ | 54.9 | 55.9 | 0.06 | ✗ | 55.1 | 56.0 | 0.06 | ✗ | 54.9 | 55.5 | 0.10 | ✗ |
| Euro-Parl | ✓ | 51.2 | 55.7 | 0.002 | ✓ | 51.5 | 55.6 | 0.004 | ✓ | 51.0 | 53.9 | 0.02 | ✓ | 50.7 | 54.2 | 0.04 | ✓ |
| | ✗ | 51.3 | 53.8 | 0.13 | ✗ | 51.3 | 53.2 | 0.31 | ✗ | 51.1 | 53.2 | 0.20 | ✗ | 51.4 | 53.3 | 0.22 | ✗ |
| Phil-Papers | ✓ | 55.3 | 59.5 | <0.001 | ✓ | 54.8 | 58.3 | <0.001 | ✓ | 54.9 | 58.0 | <0.001 | ✓ | 55.2 | 56.8 | 0.02 | ✓ |
| | ✗ | 61.2 | 55.6 | 1.00 | ✗ | 61.2 | 56.1 | 0.98 | ✗ | 61.4 | 56.2 | 0.98 | ✗ | 60.2 | 56.0 | 0.99 | ✗ |
| Hacker News | ✓ | 56.4 | 61.1 | <0.001 | ✓ | 56.5 | 59.8 | <0.001 | ✓ | 56.3 | 59.0 | <0.001 | ✓ | 56.5 | 59.1 | 0.002 | ✓ |
| | ✗ | 58.2 | 55.6 | 0.94 | ✗ | 58.2 | 57.1 | 0.24 | ✗ | 58.1 | 57.7 | 0.17 | ✗ | 58.7 | 57.7 | 0.24 | ✗ |
| Enron Emails | ✓ | 54.5 | 58.1 | <0.001 | ✓ | 54.4 | 56.9 | 0.003 | ✓ | 54.1 | 57.7 | <0.001 | ✓ | 54.4 | 56.2 | 0.04 | ✓ |
| | ✗ | 54.8 | 52.8 | 1.00 | ✗ | 54.6 | 53.0 | 0.97 | ✗ | 54.5 | 54.9 | 0.20 | ✗ | 54.6 | 54.0 | 0.76 | ✗ |
| Stack Exchange | ✓ | 54.1 | 61.9 | <0.001 | ✓ | 54.2 | 58.8 | <0.001 | ✓ | 53.9 | 58.1 | <0.001 | ✓ | 54.2 | 57.2 | 0.002 | ✓ |
| | ✗ | 52.3 | 55.0 | 0.06 | ✗ | 52.1 | 54.3 | 0.090 | ✗ | 52.1 | 54.1 | 0.24 | ✗ | 52.5 | 54.0 | 0.29 | ✗ |
| PubMed Abstract | ✓ | 54.6 | 59.7 | <0.001 | ✓ | 54.4 | 58.5 | <0.001 | ✓ | 54.1 | 58.3 | <0.001 | ✓ | 54.4 | 58.1 | <0.001 | ✓ |
| | ✗ | 54.9 | 52.6 | 1.00 | ✗ | 54.1 | 54.0 | 0.74 | ✗ | 53.9 | 53.7 | 0.83 | ✗ | 53.9 | 54.3 | 0.57 | ✗ |
| USPTO Back. | ✓ | 52.9 | 56.8 | 0.002 | ✓ | 52.8 | 56.3 | 0.004 | ✓ | 52.4 | 55.6 | 0.028 | ✓ | 52.6 | 55.4 | 0.018 | ✓ |
| | ✗ | 52.6 | 51.7 | 1.00 | ✗ | 52.9 | 52.2 | 0.99 | ✗ | 52.5 | 52.2 | 0.98 | ✗ | 52.6 | 53.1 | 0.75 | ✗ |

## G. Example of Weight Constraint

Here, we provide the following example to illustrate the importance of weight constraint. In Table A4, we show the score of three MIAs for a suspect/held-out pair on both member and non-member sets. For each suspect/held-out pair, the smaller MIA score is highlighted in **bold** in the table. We have the following observations:

1. On the *member* set, suspect data consistently shows smaller MIA scores. This occurs because membership signals have stronger effects than generation, causing suspect data to consistently yield lower MIA scores than held-out data.

2. On the *non-member* set, held-out data may exhibit smaller values for certain MIAs. This happens because generation randomness introduces fluctuation in MIA scores.

For both member and non-member sets, we train a linear model $l$ that aggregates all MIA scores to predict an overall score:

$$l(x) = \sum_i w_i \text{MIA}_i(x) \qquad (4)$$

The held-out set is labeled as 1 and the suspect set as 0. The model assigns positive weights $w_i$ to any MIA metrics $\text{MIA}_i$ on the *member* set because label $0 < 1$ and $\text{MIA}_i(\text{suspect}) < \text{MIA}_i(\text{held-out})$. However, on the *non-member* set, the model assigns a negative weight $w_3$ for $\text{MIA}_3$. This means a smaller $\text{MIA}_3$ score in the held-out set would contribute to a larger overall MIA score, which is undesirable. To address this, we constrain all weights in the linear model to be strictly positive, ensuring that a lower $\text{MIA}_i$ score can only result in a lower overall MIA score.

*Table A4.* **An example to demonstrate the importance of the weight constraint.**

| Membership | Split | Label | $\text{MIA}_1$ | $\text{MIA}_2$ | $\text{MIA}_3$ |
|:---:|:---:|:---:|:---:|:---:|:---:|
| ✓ | Suspect *(natural)* | 0 | **0.86** | **0.87** | **0.54** |
| | Held-out *(generated)* | 1 | 0.90 | 0.91 | 0.55 |
| ✗ | Suspect *(natural)* | 0 | **0.88** | **0.89** | 0.58 |
| | Held-out *(generated)* | 1 | 0.90 | 0.90 | **0.56** |

## H. Other Finetuning Configurations for Single Author Dataset

We evaluate our proposed approach on the single author dataset under different finetuning settings in Table A5. The results show that, the membership signal is stronger when the model is fine-tuned with more epochs. Also, our method performs better when full finetuning is used instead of LoRA.

*Table A5.* **Results for different fine-tuning methods.** *True* represents the true membership while *Inferred* denotes the inferred membership. Our generation is successful if these two align.

| Fine-tuning Method | True | $\text{AUC}_{\text{Text}}$ (%) | $\text{AUC}_{\text{Comb}}$ (%) | P-value | Inferred |
|:---:|:---:|:---:|:---:|:---:|:---:|
| LoRA | ✓ | 53.8 | 55.6 | 0.01 | ✓ |
| (1 epoch) | ✗ | 53.8 | 53.9 | 0.13 | ✗ |
| LoRA | ✓ | 53.7 | 56.2 | 0.005 | ✓ |
| (10 epochs) | ✗ | 53.6 | 53.5 | 0.14 | ✗ |
| Full Finetuning | ✓ | 53.7 | 56.8 | 0.008 | ✓ |
| (1 epoch) | ✗ | 53.8 | 53.7 | 0.21 | ✗ |

## I. Results on the OLMo Model

We conduct the experiments to analyze the performance with OLMo-7B model (Groeneveld et al., 2024). The OLMo-7B model is trained on the Dolma V.1.7 dataset (Soldaini et al., 2024), which has a large size of 4.5 TB. Following Duan et al. (2024), we use Dolma V.1.7 as the member set and employ Paloma (Magnusson et al., 2024) as the non-member set. The results in Table A6 demonstrate that our method successfully detects both member and non-member sets for Wikipedia and Common Crawl subsets when using the OLMo-7B model as the target model.

## J. Ablation Studies on Single Author Dataset

In addition to the ablation studies on the Pile presented in Section 5.4, we also perform the ablation studies on the single author dataset. The results in Table A7 follow a similar trend to the Pile, showing the importance of each component in our framework.

*Table A6.* **Results for OLMo-7B on different data subsets.** *True* represents the true membership while *Inferred* denotes the inferred membership. Our generation is successful if these two align.

| Subset | True | AUC $_{\text{Text}}$ (%) | AUC $_{\text{Comb}}$ (%) | P-value | Inferred |
|---|---|---|---|---|---|
| Wikipedia | ✓ | 52.9 | 55.4 | 0.009 | ✓ |
| | ✗ | 52.1 | 50.6 | 1.0 | ✗ |
| Common Crawl | ✓ | 53.5 | 55.7 | 0.01 | ✓ |
| | ✗ | 54.2 | 53.8 | 0.68 | ✗ |

*Table A7.* **Results for different configurations.** *True Membership* represents the true membership while *Inferred Membership* denotes the inferred membership. Our generation is successful if these two align.

| Configuration | True Membership | P-value | Inferred Membership |
|---|---|---|---|
| w/o Suffix Completion | ✓ | 1.0 | ✗ |
| (ICL Paraphrasing) | ✗ | 1.0 | ✗ |
| w/o Post-hoc Calibration | ✓ | <0.001 | ✓ |
| (Original T-test in DI) | ✗ | <0.001 | ✓ |
| w/o Weight | ✓ | 0.02 | ✓ |
| Constraint | ✗ | 0.08 | ✗ |
| Ours | ✓ | 0.01 | ✓ |
| | ✗ | 0.13 | ✗ |

# K. Analysis of Hyperparameter Sensitivity

We conducted a comprehensive analysis of hyperparameter sensitivity, focusing on two key parameters: the number of epochs and the number of t-test samples. The number of epochs represents the training epochs for our linear model that aggregates MIA scores. The number of t-test samples indicates the total sample size used in our statistical analysis, including both the suspect and synthetic held-out sets. Our experimental results in Table A8 demonstrate that our proposed method exhibits robust performance across a wide range of values for both hyperparameters, indicating low sensitivity to these configuration choices.

*Table A8.* **Performance of our method across different numbers of epochs and T-test samples.**

| Hyperparameter | Value | True membership | P-value | Inferred membership |
|---|---|---|---|---|
| Number of Epochs | 100 | ✓ | <0.001 | ✓ |
| | | ✗ | 1.0 | ✗ |
| | 200 | ✓ | <0.001 | ✓ |
| | | ✗ | 1.0 | ✗ |
| | 500 | ✓ | <0.001 | ✓ |
| | | ✗ | 1.0 | ✗ |
| | 1000 | ✓ | 0.003 | ✓ |
| | | ✗ | 1.0 | ✗ |
| Number of T-test Samples | 1000 | ✓ | <0.001 | ✓ |
| | | ✗ | 1.0 | ✗ |
| | 2000 | ✓ | <0.001 | ✓ |
| | | ✗ | 1.0 | ✗ |
| | 3000 | ✓ | <0.001 | ✓ |
| | | ✗ | 0.41 | ✗ |
| | 4000 | ✓ | <0.001 | ✓ |
| | | ✗ | 0.25 | ✗ |

# L. Other Related Works about Test Set Contamination Detection

Test set contamination is a newly identified risk, where the public test benchmarks are involved during LLM training (Balloccu et al., 2024). For example, Roberts et al. (2024) observe that LLMs are better at generating code with more appearances on GitHub, revealing that LLMs can be contaminated with open-source GitHub data and are overestimated on coding tasks. Similarly, Li & Flanigan (2024) demonstrate that some LLMs have a better performance on few-shot

benchmarks constructed before the model training, which indicates test set contamination for LLMs. To detect test set contamination, Golchin & Surdeanu (2023) design prompts that guide LLM to reproduce exact or near-exact test set instances, such that the model encloses the contaminated samples memorized during the pre-training phase. Oren et al. (2024) compare the target model predictions between a test set and all of its permutations. However, this method is based on the assumption that the test set is involved in the training set in its exact order, which could be interrupted by a random shuffle before training. Test set contamination can also be a potential application of our method, as the proposed approach can perform training data detection on complex datasets composed by different authors.

## M. Algorithm of Our Work

We present the detailed algorithms for our held-out data generation in Algorithm 1, and post-hoc calibration in Algorithm 2.

---

**Algorithm 1** Held-out Data Generation

---

**Require:** Documents $Doc = \{Doc_1, ..., Doc_m\}$
**Require:** Hyperparameters: Document number $m$, Maximum sequence in each document $MaxSeq$
**Ensure:** Suspect set $\mathcal{D}_{\text{sus}}$ and held-out set $\mathcal{D}_{\text{val}}$ are nearly IID
1: Initialize: $Seq, \mathcal{D}_{\text{sus}}, \mathcal{D}_{\text{val}} = \{\}, \{\}, \{\}$
2: **for** each document $doc_i \in Doc$ **do**
3:     Segment $doc_i$ into multiple sequences $\{seq_i^1, ..., seq_i^{m_i}\}$
4:     **if** $m_i < MaxSeq$ **then**
5:         $Seq_i = \{seq_i^1, ..., seq_i^{m_i}\}$
6:     **else**
7:         $Seq_i =$ randomly sampled $MaxSeq$ sequences from $\{seq_i^1, ..., seq_i^{m_i}\}$
8:     **end if**
9:     $Seq = Seq \cup Seq_i$
10: **end for**
11: Randomly split $Seq$ into generator training set $Seq_{train}$ and generator inference set $Seq_{test}$
12: Optimize generator $g$ on $Seq_{train}$ with next-token prediction loss
13: **for** each $seq_i \in Seq$ **do**
14:     $pre_i, suf_i = \text{Divide}(seq_i)$
15:     $suf_i' = g(pre_i)$
16:     $\mathcal{D}_{\text{sus}} = \mathcal{D}_{\text{sus}} \cup \{(suf_i, 0)\}$
17:     $\mathcal{D}_{\text{val}} = \mathcal{D}_{\text{val}} \cup \{(suf_i', 1)\}$
18: **end for**

---

**Algorithm 2** Post-hoc Calibration

---

**Require:** Target model $f$
**Require:** Suspect set $\mathcal{D}_{\text{sus}}$ and held-out set $\mathcal{D}_{\text{val}}$ are nearly IID.
1: Randomly split $\mathcal{D}_{\text{sus}}$ into suspect training set $\mathcal{D}_{\text{sus}}^{\text{train}}$ and suspect test set $\mathcal{D}_{\text{sus}}^{\text{test}}$
2: Randomly split $\mathcal{D}_{\text{val}}$ into held-out training set $\mathcal{D}_{\text{val}}^{\text{train}}$ and held-out test set $\mathcal{D}_{\text{val}}^{\text{test}}$
3: Optimize a text classifier $c_{text}(x)$ on $\mathcal{D}_{\text{sus}}^{\text{train}} \cup \mathcal{D}_{\text{val}}^{\text{train}}$
4: Optimize a combined classifier $c_{comb}(x, \text{MIA}(f(x)))$ on $\mathcal{D}_{\text{sus}}^{\text{train}} \cup \mathcal{D}_{\text{val}}^{\text{train}}$
5: $\mathcal{D}_{\text{text}}^{\text{diff}} = \{\}$
6: $\mathcal{D}_{\text{comb}}^{\text{diff}} = \{\}$
7: **for** $x_{\text{sus}}^{\text{test}}, x_{\text{val}}^{\text{test}} \in \mathcal{D}_{\text{sus}}^{\text{test}}, \mathcal{D}_{\text{val}}^{\text{test}}$ **do**
8:     $\mathcal{D}_{\text{comb}}^{\text{diff}} = \mathcal{D}_{\text{comb}}^{\text{diff}} \cup \{c_{comb}(x_{\text{val}}^{\text{test}}, \text{MIA}(f(x_{\text{val}}^{\text{test}}))) - c_{comb}(x_{\text{sus}}^{\text{test}}, \text{MIA}(f(x_{\text{sus}}^{\text{test}})))\}$
9:     $\mathcal{D}_{\text{text}}^{\text{diff}} = \mathcal{D}_{\text{text}}^{\text{diff}} \cup \{c_{text}(x_{\text{val}}^{\text{test}}) - c_{text}(x_{\text{sus}}^{\text{test}})\}$
10: **end for**
11: Compare and $\mathcal{D}_{\text{comb}}^{\text{diff}}$ and $\mathcal{D}_{\text{text}}^{\text{diff}}$ with t-test

---

