# OpenReview forum: "Unlocking Post-hoc Dataset Inference with Synthetic Data"
_ICML.cc/2025/Conference — ICML 2025 poster_

### Official Review · Reviewer_ZFHo · 2025-03-11

**Overall Recommendation:** 3

**Summary:**

The paper proposes a novel approach to dataset inference (DI) by addressing the challenge of requiring a held-out dataset that closely matches the suspect dataset’s distribution. The authors generate synthetic held-out data using a text generator trained on a suffix-based completion task. They further introduce a post-hoc calibration step to bridge the likelihood gaps between real and synthetic data, improving the reliability of DI. Experiments on diverse datasets demonstrate that this method enables accurate detection of whether a dataset was used in model training while minimizing false positives, making it a viable tool for copyright enforcement and data ownership verification.

**Claims And Evidence:**

I am confused by Section 4.3, and I believe more details should be added to improve the clarity of the claims.

**Essential References Not Discussed:**

I am not deeply familiar with the literature on DI, but I am well-versed in MIA research. The paper's coverage of MIA-related works is adequate.

**Experimental Designs Or Analyses:**

The experimental design has limitations due to the small model scale and dataset size. The analysis is reasonable, but the necessity of the weight constraint is not clearly demonstrated based on the ablation in Section 5.5.

**Methods And Evaluation Criteria:**

The proposed methods and evaluation criteria are reasonable (Section 5.1), but the limited model scale and dataset size constrain the generalizability of the conclusions.

**Other Comments Or Suggestions:**

**Typo**:
1. Line 283: "we rely on an important observations..."
2. Line 310: A "]" is missing

**Suggestions**:
1. I suggest giving a concrete example of MIA(f(x)), which can help readers understand the context better.
2. I strongly suggest improving the clarity of Section 4.2 and Section 4.3.

**Other Strengths And Weaknesses:**

**Strengths**:
1. The research problem is important and addresses a clear gap in existing dataset inference methods.
2. The approach is interesting, particularly the calibration step for handling distribution shifts.
3. The current experimental results are generally effective.

**Weaknesses**:
1. The experimental scale is limited, involving only small models and two datasets. The fine-tuning setup is also narrow (only one recipe is tested), limiting broader MIA evaluations. Additionally, the evaluation of text classifiers is limited, as only two specific classifiers are tested without exploring stronger alternatives.
2. The writing lacks clarity in several areas. There is a contradiction between Section 2.2’s first paragraph and Equation (1). Sections 4.2 and 4.3 are unclear, particularly the reasoning behind Equation (3) and the rationale of the weight constraint.

I believe expanding the experiments would strengthen the validation of the method.

**Questions For Authors:**

1. Can you clearly explain the relationship between $MIA(x_{val})$ and $MIA(x_{sus})$? There seems to be a contradiction in Section 2.2.
2. What is the rationale behind Equation (3)? Specifically, what is the range of values for $c(\cdot)$? Discrete or continuous?
3. Can you provide a more detailed explanation of the rationale in Section 4.3? Why does the linear regression coefficient become negative? A concrete example would be helpful.
4. Have you considered using stronger text classifiers beyond the two currently used?
5. Can your method defend against a malicious actor? Please discuss potential vulnerabilities.
6. How does your method perform under different fine-tuning recipes? My understanding is that you tested only one setting.
7. In the ablation study, you included ICL paraphrasing. I am curious about the results when using preference optimization for generation.

**Relation To Broader Scientific Literature:**

The paper addresses a key limitation of prior DI methods by eliminating the need for an in-distribution held-out dataset, making DI more practical and reliable.

**Theoretical Claims:**

The paper does not present theoretical claims, however:
1. The justification for hypothesis testing, particularly Equation (3), lacks sufficient reasoning.
2. There appears to be a contradiction between the first paragraph of Section 2.2 and Equation (1), suggesting an error in one of them.
3. The explanation in Section 4.3 is unclear.

---

> ### Author Rebuttal · Authors · 2025-04-01
>
> > **The limited model scale and dataset size constrain the generalizability of the conclusions.**
>
> We performed additional experiments on larger Pythia models (2.8B, 6.9B). Moreover, we present results on Olmo 7B, which is trained on a very large training corpus of 4.5TB. The results show our method can scale to much larger models and dataset sizes. Please refer to our response to Reviewer z1aa (https://openreview.net/forum?id=a5Kgv47d2e&noteId=z18yFkoW8x) for detailed results.
>
> > **The justification for hypothesis testing, particularly Equation (3), lacks sufficient reasoning. [...] Specifically, what is the range of values for c(⋅)? Discrete or continuous?**
>
> The difference comparison t-test in Eq. 3 is designed following our proposed suffix completion generation task. In the suffix completion task, the suspect and generated heldout suffixes have the same position within a sentence, making the two suffixes directly comparable. To this end, we propose to quantify the distributional gap in each suspect-heldout suffix pair. The gap is measured *with only text signals* (by the text classifier), and *with both text and MIA signals* (by the combined classifier). If the gaps in suspect-heldout pairs are larger with extra MIA signals, then the suspect suffixes are likely to be used for training.
>
> Specifically, the values for c(⋅) are continuous, ranging from 0 to 1.
>
> > **There appears to be a contradiction between the first paragraph of Section 2.2 and Equation (1).**
>
> We modified the text in Section 2.2 as follows:
> “The null hypothesis is that the average scores for $D_{sus}$ **are higher than or equal to** those for $D_{val}$.”
>
> > **The fine-tuning setup is narrow (only one recipe is tested)**
>
> We performed additional experiments under two more fine-tuning setups, (1) finetuning with LoRA for 10 epochs and (2) full-finetuning, as follows.
>
> | Fine-tuning Method | True | AUC Text (%) | AUC Comb (%) | P-value (Diff) | Inferred |
> |-------------------|------|--------------|--------------|----------------|----------|
> | **LoRA (1 epoch)** *(in Table 3)*  | ✓ | 53.8 | 55.6 | 0.01 | ✓ |
> | | ✗ | 53.8 | 53.9 | 0.13 | ✗ |
> | **LoRA (10 epochs)** | ✓ | 53.7 | 56.2 | 0.005 | ✓ |
> | | ✗ | 53.6 | 53.5 | 0.14 | ✗ |
> | **Full-finetuning** | ✓ | 53.7 | 56.8 | 0.008 | ✓ |
> | | ✗ | 53.8 | 53.7 | 0.21 | ✗ |
>
> Our results suggest that, with more iterations or larger parameter size, the membership signal is stronger and therefore easier to detect. This means that the setup in our submission is the most challenging one.
>
> > **Only two specific classifiers are tested without exploring stronger alternatives.**
>
> We explored a few stronger text classifier backbones and chose the simple 2-layer GPT2-based classifier.
> Considering the limited number of tokens provided by the author in the DI scenario, stronger text classifiers can be easily overfitted, especially for SoTA LLM-based text classifiers. Here, we present the results for different architectures with different parameter sizes.
>
>
> | Architecture | AUC_text (%) | Training time (minutes) |
> |-------------|------------|------------------------|
> | **GPT2 (initialized+2 layers)** | 53.3 | 0.5 |
> | **GPT2 (pretrained+lora)** | 53.0 | 26.2 |
> | **GPT2 (pretrained+full finetuning)** | 52.3 | 36.8 |
> | **Llama3-8B (pretrained+lora)** | 53.2 | 65.1 |
>
>  The results show that the simple GPT2-based classifier ((initialized+2 layers) can achieve the best AUC. Additionally, this simple classifier has a significantly shorter training time, making the method more practical when faced with more queries.
>
> > **Typo: Line 283 and Line 310**
>
> We have modified those texts in our manuscript.
>
> > **Can you provide a more detailed explanation of the rationale in Section 4.3? A concrete example would be helpful.**
>
> We added a more detailed explanation with an example in the manuscript for better understanding. We also added a figure to visually demonstrate the idea.
>
> > **Can your method defend against a malicious actor?**
>
> The existence of a malicious actor is one of the important problems we would like to address for the previous LLM Dataset Inference—the author might provide a held-out set with a distributional shift to trigger a false positive result. With our approach, **the arbitrator can synthesize a reliable held-out set** to avoid false positives caused by a malicious user. We think this should be a standard pipeline for dataset inference and visualize it in Figure 1.
>
> > **In the ablation study, you included ICL paraphrasing. I am curious about the results when using preference optimization for generation.**
>
> We extended Table 5 and provided extra results of three baselines (preference optimization, ICL text completion, ICL paraphrasing) together with two more ablation studies for each component for our generation method. Please refer to the second table in our response to reviewer hUkh (https://openreview.net/forum?id=a5Kgv47d2e&noteId=aEzlhvnQm9) for more detailed results.

---

> > ### Comment · Reviewer_ZFHo · 2025-04-06
> >
> > Thank you for your rebuttal—I have read it carefully.
> >
> > I’m generally satisfied with the new experimental results. I believe many of these should be incorporated into the final version of the paper. The original version presented rather limited empirical evidence, and these additional results (e.g., evaluations on more models and datasets, exploration of alternative classifiers, and the demonstrated ineffectiveness of other data generation methods) are important to substantiate the core claims.
> >
> > I have one remaining question regarding the experiments: why didn’t you consider using BERT or sentence-transformer as text classifiers? Also, I’m not fully convinced by your statement that “stronger text classifiers can be easily overfitted”—this sounds more like an issue with experimental implementation than a fundamental limitation.
> >
> > Regarding Equation (3), after re-reading it, I suspect there may be an issue with **the subscripts in the expectations**. I encourage you to double-check this. Please let me know if I am wrong.
> >
> > For Section 4.3, I cannot comment further since I haven’t seen the revised version.
> >
> > Overall, I find the experimental results (**after revision**) reasonably comprehensive, and the problem setting and motivation are both meaningful and interesting. **However, I still find the exposition in Sections 4.2 and 4.3—especially the methodological reasoning and notation—quite unclear.** This summarizes my view on the strengths and weaknesses of the paper. I recommend that the final decision be left to the AC’s discretion.

---

> > > ### Author Response · Authors · 2025-04-07
> > >
> > > We thank the reviewer for finding our “problem setting and motivation both meaningful and interesting”, and our “experimental results comprehensive”.
> > >
> > > > **Use BERT or sentence-transformer as text classifiers?**
> > >
> > > We also ran experiments with BERT and sentence transformer (all-MiniLM-L6-v2) as text classifiers, showing that our GPT2-based classifier overperforms both architectures.
> > >
> > > | Model | AUC_text (%) |
> > > |-------------|------------|
> > > | all-MiniLM-L6-v2 | 50.8|
> > > | BERT| 51.2|
> > > | **Ours (GPT2-based)** | 53.3 |
> > >
> > > > **“Stronger text classifiers can be easily overfitted” sounds more like an issue with experimental implementation than a fundamental limitation.**
> > >
> > > In our experimental setup where *user-provided samples are limited*, a simple GPT2-based classifier slightly outperformed other architectures. In this challenging scenario, larger classifiers likely lack sufficient data to reach their optimal performance. In real-world applications, however, an arbitrator can select the most suitable text classifier based on their specific conditions regarding data size, data type, and computation resources. This selection would give *even stronger performance* for our method.
> > >
> > > > **Subscripts in Equation 3**
> > >
> > > We modify the subscripts in Equation 3 as follows.
> > >
> > > $E_{x_{val}^{test} ∈ D_{val}^{test}, x_{sus}^{test} ∈ D_{sus}^{test}}[c_{comb}(x_{val}^{test}) - c_{comb}(x_{sus}^{test})] ≤ E_{x_{val}^{test} ∈ D_{val}^{test}, x_{sus}^{test} ∈ D_{sus}^{test}}[c_{text}(x_{val}^{test}) - c_{text}(x_{sus}^{test})].$
> > >
> > > > **I still find the exposition in Sections 4.2 and 4.3 unclear, especially the methodological reasoning and notation.**
> > >
> > > We added the following explanation regarding Sec. 4.2 and 4.3 for better understanding.
> > >
> > > **Sec. 4.2**
> > >
> > > Here we explain our dual-classifier t-test in Equation 3, where the superscripts "test" are abbreviated for simplicity.
> > >
> > > The aim of our t-test is to distinguish the distributional shift caused by *membership signal* from the shift caused by *generation*. These two kinds of distributional shifts exist between every suspect suffix $x_{sus} ∈ D_{sus}$ and its generated held-out version $x_{val} ∈ D_{val}$.
> > >
> > > We train two classifiers to achieve this goal, as described in Sec. 4.2, lines 283-307. Here, a classifier $c$ predicts a probability $c(x)$ (ranging from 0 to 1) that the input $x$ is from the held-out set $D_{val}$. Therefore, the difference in the predicted score $c(x_{val}) - c(x_{sus})$ shows how well the classifier can distinguish the suspect-heldout pair $x_{val}$ and $ x_{sus}$.  In every such pair, we quantify the shift caused by *generation* with the text classifier as $c_{text}(x_{val}) - c_{text}(x_{sus})$. We also quantify the combined effects caused by *generation and membership* with the combined classifier as $c_{comb}(x_{val}) - c_{comb}(x_{sus})$.
> > >
> > > If the membership signal is present, the combined effects will be stronger than the generation effect alone, and the predicted probability will be slightly more accurate for the combined classifier, i.e. $c_{comb}(x_{val}) - c_{comb}(x_{sus}) > c_{text}(x_{val}) - c_{text}(x_{sus})$. This will be examined with a t-test, where the null hypothesis is formalized as in Equation 3.
> > >
> > > **Sec. 4.3**
> > >
> > > We illustrate our approach with the following example. For each suspect-heldout pair, the smaller MIA score is highlighted in bold in the table. We observe:
> > >
> > > 1) On the **member set**, suspect data consistently shows smaller MIA scores. This occurs because membership signals have stronger effects than generation, causing suspect data to consistently yield lower MIA scores than held-out data.
> > >
> > > 2) On the **non-member set**, held-out data may exhibit smaller values for certain metrics. This happens because generation randomness introduces fluctuation in MIA scores.
> > >
> > > For both member and non-member sets, we train a linear model $l$ that aggregates all MIA scores to predict an overall score:
> > >
> > > $l(x) = \sum_i w_i MIA_i(x)$
> > >
> > > The held-out set is labeled as 1 and the suspect set as 0. The model assigns positive weights $w_i$ to any MIA metrics $MIA_i$ on the *member set* because label 0<1 and $MIA_i($suspect$)<MIA_i($held-out$)$. However, on the *non-member set*, the model assigns a negative weight $w_3$ for $MIA_3$. This means a smaller $MIA_3$ score in the held-out set would contribute to a larger overall MIA score, which is undesirable. To address this, we constrain all weights in the linear model to be strictly positive, ensuring that a lower $MIA_i$ score can only result in a lower overall MIA score.
> > >
> > > | Membership | Split | Label | $MIA_1$ | $MIA_2$ | $MIA_3$ |
> > > |------------|------------|---------------------|-----------------------------------|-----------------------------------|----------------------------------------|
> > > | ✓ | Suspect (*natural*) | 0 | **0.86** | **0.87** | **0.54** |
> > > | | Held-out (*generated*) | 1 | 0.90 | 0.91 | 0.55 |
> > > | ✗ | Suspect (*natural*) | 0 | **0.88** | **0.89** | 0.58 |
> > > | | Held-out (*generated*) | 1 | 0.90 | 0.90 | **0.56** |

---

### Official Review · Reviewer_hUkh · 2025-03-13

**Overall Recommendation:** 3

**Summary:**

The work presents a method to generate synthetic data for use with Dataset Inference (DI) algorithms, which require held-out examples from the dataset distribution of interest. DI algorithms are used to detect the presence of certain data distributions within the training data of a particular model.  The authors argue that a (non-synthetic) held out dataset is rarely available and that even small shifts in the distribution of a held out set can cause DI algorithms to fail. Furthermore, the authors define a "post-hoc calibrated" test procedure, to use in conjunction with this synthetic dataset, to identify when a suspected dataset has been included in a model's training set.

**Claims And Evidence:**

The submission claims two contributions, the procedure for generating synthetic held-out data and a second procedure for testing whether the suspect data was included in the target model's training set via a "post-hoc calibration".  The impact of the post-hoc calibration seems quite well supported by the empirical evaluation of Section 5. The necessity for a more complex data generation procedure is somewhat supported in Table 5, but it is unclear from the text, is this result only for the single-author blogpost dataset? Do similar trends also hold for datasets from the Pile?

**Essential References Not Discussed:**

N/A

**Experimental Designs Or Analyses:**

The empirical methodology seems sound (apart from my comments related to methods and evaluation criteria).

**Methods And Evaluation Criteria:**

The benchmark datasets (The Pile and blog posts) are reasonable, however, the authors do not mention what blog posts they use.  Could this be clarified?

The model (Pythia 410M) is relatively small so there is some question as to how well the method performs with model scale.  Additional model sizes (2B, 7B) would help illustrate the scaling behavior.

Another simple data generation baseline, which should be considered, is an in-context suffix completion approach. One where you prompt the model to complete a suffix (selected in the same fashion as done for the proposed suffix completion approach). Furthermore, a few full examples could be included in the context to give the model some guidance on the writing style.

**Other Comments Or Suggestions:**

- As motivated by sec 3.3, using a bad (i.e. out of distribution) held-out set can create false positives...this begs the question, should we use distribution testing to ensure that the suspect and held out set are statistically similar as a standard part of the DI protocol?

-  "The null hypothesis is that the average scores for D_sus is lower than for D_val"
Shouldn't the null hypothesis (eq 1) be that the scores for the two distributions are the same, i.e. in the case that D_sus and D_val are drawn from the same distribution and neither has been included in the training set?  If we find D_sus has a much higher score, then we reject the null and conclude D_sus was included in the training set.

- Please make clear earlier that larger MIA score implies the datapoint is included in the training set. At least, that is what I inferred from how MIA is being used in Eq (1).

- Figure 3, left-panel, step 2: all the text snippets in the illustration have identical text. Is this intended?

- This is minor, but I found the use of red checks to indicate positive and green/blue crosses to indicate negative confusing, since I usually think of red to indicate negative and green to indicate positive. I suggest using green checks and red crosses.

- Does the inferred column in Table 4 need to be updated to indicate some of the errors, for example in the case of Ubuntu IRC it looks like there is a false positive?

**Other Strengths And Weaknesses:**

As presented, the paper presents two main contributions -- (1) how to generate synthetic data for solving the DI problem and (2) how to conduct tests that take into account the synthetic data.  The paper could be strengthened and made more clear, if there are additional experiments to support (1), i.e., how critical is the synthetic data generation approach given (2).  If it is indeed critical, it would be great to have further empirical support and if it is not important (e.g. any reasonable data generation approach works well when coupled with (2)), then the paper should be updated to emphasize (2) more, in my opinion.

**Questions For Authors:**

(1) Can you provide some (even preliminary) empirical results for using an in-context suffix completion approach (as suggested in the Experimental Design section above)?  This would serve as an ablation to measure the effectiveness of suffix completion with LoRA vs using in-context instructions.

(2) Can you provide additional experiments showing that the synthetic generation method is necessary (and that the post-hoc calibration alone is not sufficient)? Essentially, expanding the result of Table 5.

**Relation To Broader Scientific Literature:**

The paper does a reasonable job of introducing DI concepts and related works.

**Theoretical Claims:**

No theoretical claims were made.

---

> ### Author Rebuttal · Authors · 2025-04-01
>
> > **Is the result in table 5 only for the single-author dataset? Do similar trends also hold for the Pile dataset?**
>
> The results in Table 5 are for the Pile. Below, we also show that our data generation procedure is necessary for the single-author dataset.
>
> | Configuration | True Membership | P-value | Inferred Membership |
> |--------------|----------------|---------|---------------------|
> | **w/o Suffix Completion (ICL Paraphrasing)** | ✓ | 1.0 | ✗ |
> | | ✗ | 1.0 | ✗ |
> | **w/o Post-hoc Calibration (Original T-test in DI)** | ✓ | <0.001 | ✓ |
> | | ✗ | <0.001 | ✓ |
> | **w/o Weight Constraint** | ✓ | 0.02 | ✓ |
> | | ✗ | 0.08 | ✗ |
> | **Ours** | ✓ | 0.01 | ✓ |
> | | ✗ | 0.13 | ✗ |
>
> The results for single-author data follow a similar trend as the Pile, showing the importance of each component in our framework.
>
> >**The authors do not mention what blog posts they use. Could this be clarified?**
>
> The blog posts are from the financial domain, regarding stock investment suggestions. Due to copyright reasons, the blog post data is not available for sharing yet.
>
> >**Additional model sizes (2B, 7B) would help illustrate the scaling behavior.**
>
> We provide the results to demonstrate the effectiveness of our method across larger model sizes of Pythia (2.8B and 6.9B) and on different model architecture (Olmo 7B). Please refer to our response to Reviewer z1aa (https://openreview.net/forum?id=a5Kgv47d2e&noteId=z18yFkoW8x) for detailed results.
>
> >**Another simple data generation baseline, which should be considered, is an in-context suffix completion approach.**
>
> We performed this baseline in Appendix A.1 “Text Completion Prompt”. Concretely, as an approach of in-context learning, we give ten examples from the documents, and prompt the GPT-4-Turbo model to finish an incomplete document. This baseline performs slightly worse than ICL paraphrasing, and the GPT2 classifier can easily detect the distributional shift with an AUC of 99.2%.
>
> >  **How critical is the synthetic data generation approach given contribution (2)?**
>
> Besides the results in Table 2 and Appendix A, we also extended the following results in Table 5 to show the importance of our data generation approach. Concretely, we show that all the generation baselines do not work even with our self-calibration methods (Row 1-6). Moreover, we show that each component in our generation procedure is critical to our framework (row 7-10).
>
> | Configuration | True Membership | P-value | Inferred Membership |
> |--------------|----------------|---------|---------------------|
> | **w/o Suffix Completion (ICL Text Completion)** | ✓ | 1.0 | ✗ |
> | | ✗ | 1.0 | ✗ |
> | **w/o Suffix Completion (ICL Paraphrasing)** | ✓ | 1.0 | ✗ |
> | | ✗ | 1.0 | ✗ |
> | **w/o Suffix Completion (Performance Optimization)** | ✓ | 1.0 | ✗ |
> | | ✗ | 1.0 | ✗ |
> | **w/o segment and shuffle** | ✓ | 1.0 | ✗ |
> | | ✗ | 1.0 | ✗ |
> | **w/o suffix completion** | ✓ | 1.0 | ✗ |
> | | ✗ | 1.0 | ✗ |
> | **Ours** | ✓ | <0.001 | ✓ |
> | | ✗ | 1.0 | ✗ |
>
>
>
> > **should we use distribution testing to ensure that the suspect and held-out set are statistically similar as a standard part of the DI protocol?**
>
>
> Adding distributional testing as part of DI protocol only strengthens its *robustness* but not *utility*. When a distributional shift is identified, it is still hard for the author to provide held-out data required for DI (as shown in Sec. 3). This disables further DI procedures, even when there is a copyright infringement. This is exactly the problem we would like to address in this work—instead of asking the author to provide statistically similar held-out data, **the arbitrator should be able to synthesize a reliable held-out set**. We think this should be a standard protocol for dataset inference and visualize it in Figure 1.
>
> > **"The null hypothesis is that the average scores for $D_{sus}$ is lower than for $D_{val}$" Shouldn't the null hypothesis (eq 1) be that the scores for the two distributions are the same?**
>
> The text should be “The **alternative** hypothesis is that the average scores for $D_{sus}$ is lower than for $D_{val}$”. If $D_{sus}$ is trained on the target model, then its MIA scores (e.g. loss) should be lower than the IID held-out set $D_{val}$. The null hypothesis, against the alternative hypothesis, is that “the average scores for $D_{sus}$ is higher than or the same as those for $D_{val}$”. This also aligns with Eq. 1.
>
> >**Does the inferred column in Table 4 need to be updated to indicate some of the errors, for example in the case of Ubuntu IRC it looks like there is a false positive?**
>
> As pointed out in Sec. 5.3, although the **difference comparison** t-test works for all subsets, the **AUC comparison** t-test fails for three subsets: Ubuntu IRC, PubMed Central, and PhilPapers. Therefore, we suggest the difference comparison t-test as a better metric. We have also modified the table accordingly.
>
> > **Suggestions for figures and symbols**
>
> We thank the suggestions from the reviewer, and made changes as suggested.

---

> > ### Comment · Reviewer_hUkh · 2025-04-02
> >
> > The author's response effectively addressed some of my previous concerns, leading me to increase my score to a 3.

---

> > > ### Author Response · Authors · 2025-04-03
> > >
> > > We appreciate your time and effort in reviewing our work. Thank you for your thoughtful feedback and for reconsidering our submission. We are happy that our responses addressed the concerns. We would be grateful for any additional guidance on potential further improvements.

---

### Official Review · Reviewer_z1aa · 2025-03-14

**Overall Recommendation:** 4

**Summary:**

This paper presents a method for Unlocking Post-hoc Dataset Inference (DI) with Synthetic Data, for safeguarding intellectual property in the era of Large Language Models. The authors claim that synthetic data generation combined with post-hoc calibration can robustly enable DI, allowing data owners to verify unauthorized data usage in LLM training.

**Claims And Evidence:**

The claims made in the submission are well-supported by experimental evidence.

**Essential References Not Discussed:**

To the best of my knowledge, there are no essential references that are not discussed.

**Experimental Designs Or Analyses:**

The authors evaluate their method on single-author blog posts and subsets of the Pile dataset, which include diverse text types (e.g., Wikipedia, ArXiv, GitHub). These datasets are representative of real-world scenarios where DI might be applied, such as copyright claims for blog posts or academic papers.

**Methods And Evaluation Criteria:**

The proposed methods and evaluation criteria are well-designed and appropriate for the problem of post-hoc dataset inference. The authors provide clear and convincing evidence that their approach effectively addresses the challenges of DI.

**Other Comments Or Suggestions:**

No other comments.

**Other Strengths And Weaknesses:**

Strengths:
The paper is well-written and contains extensive experimental results.
It touches on an important topic of safeguarding intellectual property against Large Language Models.

Weaknesses:
The method is evaluated on single-author blog posts and subsets of the Pile dataset, which are relatively homogeneous or well-structured. However, it may struggle with highly specialized datasets (e.g., medical, legal, or technical texts) where domain-specific patterns might not be captured. This could lead to larger distributional shifts and reduced reliability in these contexts.

**Questions For Authors:**

No other questions.

**Relation To Broader Scientific Literature:**

Previous work on Dataset Inference by Maini et al. (2021) provides a method to determine whether a suspect dataset was used to train a machine-learning model. However, DI traditionally requires an in-distribution held-out dataset, which is rarely available in practice. The paper addresses the critical limitation of DI by proposing a method to synthetically generate in-distribution held-out data.

**Theoretical Claims:**

All theoretical claims have been checked and seem valid.

---

> ### Author Rebuttal · Authors · 2025-03-31
>
> We thank the Reviewer for the positive feedback and encouraging comments.
>
> >**However, it may struggle with highly specialized datasets (e.g., medical, legal, or technical texts) where domain-specific patterns might not be captured. This could lead to larger distributional shifts and reduced reliability in these contexts.**
>
> We would like to note that the Pile is not a homogeneous dataset and has many different data subsets from various domains, formats, and languages. For example, the PubMed Central (PMC) is a subset of the PubMed online repository for biomedical articles (*medical*). The FreeLaw subset contains opinions from federal and state courts (*legal*). The ArXiv subset contains *technical text* mostly from papers in the fields of Math, Computer Science, and Physics. EuroParl is a *multilinguall* corpus consisting of the proceedings of the European Parliament in 21 European languages. StackExchange contains user-posted content on the Stack Exchange network in the *question-answer format* covering a wide range of subjects. We present examples of these synthetic texts from the different subsets of the Pile in the Appendix E. These samples show the diversity of the subsets in the Pile. Here, we also demonstrate that our method generalizes well across different pile subsets and model parameter sizes.
>
> We further extend our analysis beyond the Pile and test our method on the Dolma dataset, which was built from a diverse mixture of web content, scientific papers, code, public-domain books, social media, and encyclopedic materials. We run additional experiments using Olmo 7B, with the member data from Dolma-v1_7 and the non-member data from Paloma. We observe that our method is also effective for the different data types. The results are provided in the table below (following Table 4 in our submission):
>
> | **Model** | **Subset** | **True** | **AUC Text (%)** | **AUC Comb (%)** | **P-value (Diff)** | **Inferred** |
> |-------|--------|------|--------------|--------------|----------------|----------|
> | **Pythia-2.8B** | ArXiv | ✓ | 53.0 | 59.1 | <0.001 | ✓ |
> |  |  | ✗ | 53.1 | 53.3 | 0.84 | ✗ |
> |  | StackExchange | ✓ | 53.7 | 57.7 | <0.001 | ✓ |
> |  |  | ✗ | 52.3 | 53.7 | 0.18 | ✗ |
> |  | EuroParl | ✓ | 50.5 | 54.0 | 0.009 | ✓ |
> |  |  | ✗ | 51.1 | 52.3 | 0.17 | ✗ |
> | **Pythia-6.9B** | ArXiv | ✓ | 53.6 | 60.0 | <0.001 | ✓ |
> |  |  | ✗ | 53.5 | 54.1 | 0.37 | ✗ |
> |  | StackExchange | ✓ | 53.7 | 58.4 | <0.001 | ✓ |
> |  |  | ✗ | 52.4 | 53.1 | 0.21 | ✗ |
> |  | EuroParl | ✓ | 50.8 | 55.8 | <0.001 | ✓ |
> |  |  | ✗ | 50.7 | 53.4 | 0.11 | ✗ |
> | **Olmo-7B** | Wikipedia | ✓ | 52.9 | 55.4 | 0.009 | ✓ |
> |  |  | ✗ | 52.1 | 50.6 | 1.0 | ✗ |
> |  | Common Crawl | ✓ | 53.5 | 55.7 | 0.01 | ✓ |
> |  |  | ✗ | 54.2 | 53.8 | 0.68 | ✗ |

---

### Official Review · Reviewer_thbe · 2025-03-15

**Overall Recommendation:** 1

**Summary:**

This paper introduces a framework for post-hoc dataset inference in large language models (LLMs) by synthesizing held-out data. The central motivation is to address the critical bottleneck of conventional dataset inference methods, which require an in-distribution held-out set that is rarely available in practice. To overcome this limitation, the authors propose generating synthetic held-out data using a fine-tuned data generator trained on a suffix completion task, designed to mimic the distribution of the suspect dataset. Since synthetic data may introduce distributional shifts relative to natural data, the paper further introduces a post-hoc calibration step that leverages a dual-classifier approach—one classifier distinguishes natural versus synthetic text, while a membership-aware classifier combines textual features with membership signals (e.g., perplexity and token probabilities). The method is evaluated on both single-author datasets (such as blog posts) and larger, heterogeneous collections (e.g., various subsets from the Pile), demonstrating that the synthetic held-out data, once calibrated, can reliably reveal whether a suspect dataset was used in LLM training. The authors also present ablation studies comparing different generation strategies (including in-context paraphrasing, preference optimization, and suffix completion), showing that suffix completion yields the smallest distributional gap.

**Claims And Evidence:**

1. The paper claims to unlock reliable post-hoc dataset inference by synthetically generating an in-distribution held-out set that can serve as a surrogate for real data.
2. The authors designed data generation pipeline based on a suffix completion task, followed by a post-hoc calibration procedure using a dual-classifier approach.

**Essential References Not Discussed:**

Incorporating a comparison with more recent methods for LLM membership inference could strengthen the literature review.

**Experimental Designs Or Analyses:**

Experiments are conducted on both a single-author blog dataset and multiple subsets from the Pile dataset. The target models are LLMs (e.g., fine-tuned Pythia 410M) and the synthetic data is generated via multiple strategies, with suffix completion proving most effective.
However, the evaluation might be strengthened by testing on more recent LLM architectures and diverse model sizes.

**Methods And Evaluation Criteria:**

The methodology focuses on generating synthetic held-out data from the suspect set using a fine-tuned generator and then aligning the generated and natural data distributions via post-hoc calibration. The dual-classifier setup—combining a text-only classifier with a membership-aware classifier—is used to disentangle genuine membership signals from artifacts caused by distributional shifts.

**Other Comments Or Suggestions:**

Refer to weaknesses

**Other Strengths And Weaknesses:**

Strengths:
1. The paper addresses a practical bottleneck in dataset inference by eliminating the need for a reserved held-out set.
2. The paper is easy to follow.

Weaknesses:
1. The approach for synthesizing held-out data, using techniques like suffix completion and preference optimization—does not represent a significant departure from existing methods for synthetic data generation. Similar strategies have been explored in other works.
2. While the authors claim that their dual-classifier t-test framework uniquely disentangles genuine membership signals from mere distribution shifts, its novelty compared to the approach used in “LLM Dataset Inference: Did you train on my dataset?” remains unclear.
3. Sensitivity to different LLM architectures and hyperparameter settings is not fully explored.

**Questions For Authors:**

1. How does your approach for synthesizing held-out data—using prompted paraphrasing, suffix completion and preference optimization—improve upon or differ from prior methods for synthetic data generation?
2. Can you clarify the unique aspects of your dual-classifier t-test framework compared to the methodology presented in “LLM Dataset Inference: Did you train on my dataset?”
3. What is the sensitivity analyses on various LLM architectures and hyperparameter settings?

**Relation To Broader Scientific Literature:**

The work is well positioned within the literature on dataset inference, membership inference attacks, and test set contamination.

**Theoretical Claims:**

The dual-classifier and t-test framework are theoretically motivated by prior work in membership inference attacks.

---

> ### Author Rebuttal · Authors · 2025-03-31
>
> >**The evaluation might be strengthened by testing on more recent LLM architectures and diverse model sizes.**
>
> We provide results to demonstrate the effectiveness of our method across larger model sizes of Pythia (2.8B and 6.9B) and different model architecture (Olmo 7B). Please refer to our response to Reviewer z1aa (https://openreview.net/forum?id=a5Kgv47d2e&noteId=z18yFkoW8x) for detailed results.
>
>
> >**Weakness 1 & Question 1: How does your approach for synthesizing held-out data [...] differ from prior methods for synthetic data generation?**
>
> The goal of our work is not to propose a novel method for synthetic data generation. Instead, our main contributions lie in combining *high-quality and nearly IID synthetic data generation* with *post-hoc calibration* within a framework of dataset inference to provide a practical tool for real-world litigations.
>
> Moreover, we note that directly applying previous generation methods only have limited performance for our task (shown in Table 2). Therefore, we introduce two simple but effective designs tailored for our framework:
>
> (1) **Segmenting and shuffling** author-provided documents: The segmented and shuffled sequences constructs IID generator training and inference splits, facilitating easier synthesis by the data generator.
>
> (2) **Suffix comparison** scheme: Both the suspect and synthetic suffixes share a common prefix, which ensures the two suffixes have the same position within a sentence. This enables direct comparison between original-synthetic suffix pairs, which also supports the difference comparison t-test in our *post-hoc calibration*.
>
> We present the following results to demonstrate the importance of the two key designs in our held-out data synthesis:
>
>
> | **Approach** | **True Membership** | **AUC_Text (%)** | **AUC_Comb (%)** | **P-value** | **Inferred Membership** |
> |----------------------|-------------------|----------------|-----------------|-------------|------------------------|
> | **w/o segment and shuffle** | ✓ | 72.5 | 45.8 | 1.0 | ✗ |
> |  | ✗ | 72.6 | 42.2 | 1.0 | ✗ |
> | **w/o suffix comparison** | ✓ | 62.8 | 52.9 | 1.0 | ✗ |
> | | ✗ | 62.4 | 50.3 | 1.0 | ✗ |
> | **Ours (w/ segment and shuffle+suffix comparison)** | ✓ | 53.6 | 59.7 | <0.001 | ✓ |
> |  | ✗ | 53.1 | 46.6 | 1.0 | ✗ |
>
>
> With the two proposed designs, our method has the lowest AUC_Text, indicating it can synthesize nearly IID held-out data. Moreover, the generated held-out set has a significantly higher AUC_Comb on the member set than on the non-member set, which shows our synthetic texts are effective for detecting membership signals.
>
> >**Weakness 2 & Question 2: Clarify the unique aspects of your dual-classifier t-test framework compared to the method in “LLM Dataset Inference”**
>
> In Section 3, we show that the “LLM Dataset Inference” fails since it uses *the simple null hypothesis* that “the suspect dataset was not used for training” with a t-test that compares the MIA scores between *the suspect and the held-out sets*.
>
> By introducing *statistical tests based on dual-classifier*, we can statistically distinguish distributional shifts caused by actual membership signals from those caused by generation, and provide more robust detection results. Specifically, our method compares the shifts between suspect and held-out sets *with and without MIA metrics*, which are measured by the text classifier and the combined classifier, respectively.
>
> >**Weakness 3 & Question 3: Sensitivity analysis on hyperparameter settings**
>
>
> In our proposed framework, the arbitrator, who verifies the training data usage, can choose hyperparameters in both the held-out data generation and post-hoc calibration stages.
>
> In the **held-out data generation** stage, the arbitrator can choose hyperparameters according to the AUC of the text classifier, where a low AUC_text indicates nearly IID suspect and synthetic held-out texts.
>
> In the **post-hoc calibration** stage, there are a few important hyperparameters for our method, especially epochs for training the linear classifiers and t-test sample sizes. We present the sensitivity analysis of our method for these hyperparameters below (we use 200 epochs and 2000 sample size in our manuscript).
>
> | **Hyperparameter** | **Value** | **True membership** | **P-value (diff)** | **Inferred membership** |
> |----------------|-------|-----------------|----------------|---------------------|
> | **Epoch** | 100 | ✓ | <0.001 | ✓ |
> |  |  | ✗ | 1.0 | ✗ |
> |  | **200** | ✓ | <0.001 | ✓ |
> |  |  | ✗ | 1.0 | ✗ |
> |  | 500 | ✓ | <0.001 | ✓ |
> |  |  | ✗ | 1.0 | ✗ |
> |  | 1000 | ✓ | 0.003 | ✓ |
> |  |  | ✗ | 1.0 | ✗ |
> | **Num. t-test samples** | 1000 | ✓ | <0.001 | ✓ |
> |  |  | ✗ | 1.0 | ✗ |
> |  | **2000** | ✓ | <0.001 | ✓ |
> |  |  | ✗ | 1.0 | ✗ |
> |  | 3000 | ✓ | <0.001 | ✓ |
> |  |  | ✗ | 0.41 | ✗ |
> |  | 4000 | ✓ | <0.001 | ✓ |
> |  |  | ✗ | 0.25 | ✗ |
>
> The results show that our method can tolerate a wide range of hyperparameters and gives true positives and true negatives.

---

### Decision · Program_Chairs · 2025-05-01

**Decision:**

Accept (poster)

**Comment:**

This paper presents a framework for dataset inference (DI) in large language models, addressing the practical limitation of needing a real in-distribution held-out set. The proposed approach combines suffix-based synthetic data generation with a post-hoc calibration procedure using a dual-classifier t-test.

After the rebuttal, three reviewers recommended acceptance, citing the clarity of the presentation, the importance of the problem, and the expanded experimental scope. One reviewer remained negative, raising three points: (i) limited novelty in the data‑synthesis component, (ii) missing hyper‑parameter ablations, and (iii) insufficient differentiation from a related paper. The authors’ rebuttal satisfactorily addressed points (ii) and (iii); the meta‑reviewer further agreed with the authors' argument that the work’s goal is to harness - not advance - data‑synthesis methods for DI, which mitigates concern (i).

Overall, the framework is a meaningful contribution to practical DI scenarios, though the authors are encouraged to further improve the clarity of the methodological exposition as recommended by one of the reviewers.